# Altered thymic niche synergistically drives the massive proliferation of malignant thymocytes

**Erika Tsingos[1]\*[†], Advaita M Dick[2][†], Baubak Bajoghli[2]\*[‡]**

[1]Computational Developmental Biology Group, Institute of Biodynamics and Biocomplexity, Utrecht University, Utrecht, Netherlands; [2]Department of Hematology, Oncology, Immunology, and Rheumatology, University, Hospital of Tübingen, Tübingen, Germany

**\*For correspondence:**
e.tsingos@uu.nl (ET);
bajoghli@austrian-bioimaging.at (BB)

[†]These authors contributed equally to this work

**Present address:** [‡]Austrian BioImaging/CMI, Vienna, Austria

## eLife Assessment

This **important** study combines agent-based modelling and in vivo experiments in medaka embryos to provide new insights into the role of the thymic niche in T cell development. The modelling yields some interesting and **solid** findings regarding the importance of thymic epithelial cells. This study would be of interest to oncologists, immunologists, and mathematical modelers.

**Abstract** The discovery of genetic alterations in patient samples over the last decades has reinforced a cell-autonomous view of proliferative expansion during T-cell acute lymphoblastic leukemia (T-ALL) development in the thymus. However, the potential contribution of non-cell-autonomous factors, particularly the impact of thymic epithelial cells (TECs) within the thymic niche during the initiation phase, remains unexplored. In this study, we combine a cell-based computational model of the thymus with complementary in vivo experiments in medaka (*Oryzias latipes*) to systematically analyze the impact of 12 cell-autonomous and non-autonomous factors, individually and in combination, on the proliferation of normal and malignant thymocytes carrying interleukin-7 receptor (IL7R) gain-of-function mutations or elevated IL7R levels, as observed in T-ALL patients. By simulating over 1500 scenarios, we show that while a dense TEC network favored the proliferation of normal thymocytes, it inhibited the proliferation of malignant lineages, which achieved their maximal proliferative capacity when TECs were sparsely distributed. Our in silico model further predicts that specific mutations could accelerate proliferative expansion within a few days. This prediction was experimentally validated, revealing the rapid onset of thymic lymphoma and systemic infiltration of malignant T cells within just 8 days of embryonic development. These findings demonstrate that synergistic interaction between oncogenic alterations and modifications in the thymic niche can significantly accelerate disease progression. Our results also suggest that negative feedback from the proliferative state suppresses thymocyte differentiation. Overall, this multidisciplinary work reveals the critical role of TEC-thymocyte interactions in both the initiation and progression of T-ALL, highlighting the importance of the thymic microenvironment in early leukemogenesis.

## Introduction

T-cell acute lymphoblastic leukemia (T-ALL) is a malignancy characterized by the proliferation of immature T cells, known as thymocytes, and is subclassified according to the stage of thymic maturation (*Terwilliger and Abdul-Hay, 2017*; *De Keersmaecker et al., 2013*). Representing approximately 10–15% of pediatric cases and 25% of adult cases of all acute lymphoblastic leukemia (*Vadillo et al.,*

*2018*), T-ALL originates in the thymus. This organ is the primary site where early T-cell progenitors (ETPs) originating from the hematopoietic tissue differentiate into mature, immunocompetent T cells. Thymocytes follow a well-orchestrated migratory path within distinct thymic niches, receiving essential extrinsic signals for their proliferation, differentiation, and selection (*Takahama, 2006*). Thymic epithelial cells (TECs) play a central role in this process, acting as the main cellular component of these niches, and influencing thymocyte migration and differentiation by expressing crucial growth factors, ligands (such as DLL4), or releasing chemokines (e.g. CCL25) and cytokines such as interleukin-7 (IL-7) into the thymic niche (*Yui and Rothenberg, 2014*; *Koch et al., 2008*; *Zamisch et al., 2005*).

The etiology of T-ALL is a topic of extensive research. Isolation and genome sequencing of malignant thymocytes from patients have uncovered mutations in over 100 putative driver genes (*Liu et al., 2017*; *Girardi et al., 2017*). Notably, over 50% of T-ALL patients harbor gain-of-function mutations in the NOTCH1 gene, which encodes the receptor for DLL4, while around 10% of patients exhibit similar mutations in the *IL-7 receptor* (*IL7R*) gene (*Liu et al., 2017*). The leukemogenic potential of constitutive activation of these receptors has been demonstrated across various vertebrate models (*González-García et al., 2009*; *Silva et al., 2021*; *Shochat et al., 2011*; *Zenatti et al., 2011*; *Oliveira et al., 2022*; *Chen et al., 2007*; *Blackburn et al., 2012*), emphasizing the evolutionary conservation of thymic developmental mechanisms that become misrouted in T-ALL (*Bajoghli et al., 2019*). These studies have contributed valuable insights into the impact of genetic alterations and oncogenic pathways on T-ALL development. However, the gene-centric view provided only limited understanding of the initiation and progression of the disease. In recent years, the growing evidence of the role of an abnormal tumor microenvironment in carcinogenesis strongly supports a more integrative view that considers the convergence of cellular genetics and the surrounding malignant niche as crucial elements for disease progression (*Vadillo et al., 2018*). In the context of T-ALL, however, the impact of the thymic niche or the role of TECs has not been studied.

Computational and mathematical models are powerful tools for integrating complex biological processes and can be used to test new hypotheses in health and disease (*Ji et al., 2017*; *Metzcar et al., 2019*; *King et al., 2021*). Several such models have been developed to study population dynamics during T-cell development (*Robert et al., 2021*; *Aghaallaei et al., 2021*; *Efroni et al., 2007*; *Thomas-Vaslin et al., 2008*; *Vibert and Thomas-Vaslin, 2017*; *Souza-e-Silva et al., 2009*). Our recently developed cell-based computational model, for example, simulates individual thymocytes using agents within a spatially resolved 'virtual thymus' to explore cell-level behaviors and their effects on thymocyte population dynamics. This model was originally developed to investigate the impact of thymic niche signals and intrathymic cell localization on the αβ/γδ T-cell sublineage outcomes (*Aghaallaei et al., 2021*).

In this study, we developed new computational tools to enhance the virtual thymus model, enabling us to identify both cell-autonomous and non-autonomous factors in thymocytes and TECs that could drive the proliferative expansion of a lineage derived from a single progenitor cell (hereafter referred to as a clone) within the thymus. We conducted an unbiased systematic analysis of various parameters, including TEC shape and architecture, IL-7 signaling, cell cycle, duration of the proliferative phase, and cell migration. This analysis facilitated a detailed comparison of proliferative expansion between wild-type (WT) and lesioned clones with alterations in the IL7R and NOTCH1 receptors. Through simulations of over 1500 scenarios, we identified factors that promote substantial expansion of malignant clones. In particular, modifications in the TEC network and niche emerged as a previously uncharacterized factor that could synergistically accelerate clonal expansion. Subsequent in vivo experiments using medaka fish (*Oryzias latipes*) confirmed the outcomes of our in silico analysis, solidifying the predictive accuracy of our virtual model and revealing the impact of TECs on the initiation and progression of T-ALL.

## Results and discussion
### The virtual thymus model enables reconstructing clonal dynamics of medaka embryonic T-cell development

The details of the virtual thymus model implementation and parameter estimation have been extensively described previously (see Supplementary Material of *Aghaallaei et al., 2021*), and in the following, we will only delineate its main features. The model considers both spatial and temporal

aspects of early thymus development at multiple scales: Cells are physically represented as one or multiple spheres that mechanically interact with adhesive and repulsive forces to capture cell crowding; each cell has an independent internal state that changes dynamically based on subcellular signaling modeled with ordinary differential equations or with phenomenological rules; and at the tissue level, a partial differential equation is used to solve the diffusion of extracellular molecules such as cytokines. To reduce computational cost, the computational representation focuses on a 5 µm deep slice of the lower half of the radially symmetric organ, approximately 1/10 of the total volume of a medaka embryonic thymus (*Figure 1—figure supplement 1A and B*). Consistent with observations in vertebrates, including in medaka (*Bajoghli et al., 2009*; *Bajoghli et al., 2015*; *Aghaallaei et al., 2021*; *Aghaallaei et al., 2022*), ETPs within our virtual thymus mechanically and biochemically interact with TECs, proliferate and differentiate in response to signals from the thymic niche, and subsequently diverge into two distinct T-cell sublineages before selection and exit from the thymus (*Figure 1A*; see Appendix 1 and *Aghaallaei et al., 2021*, for more technical details). In this work, we define cells of a lineage derived from a single founder ETP as a 'clone', and we do not consider differences in immunological clonotypes that emerge after recombination of the T-cell receptor.

In the virtual thymus model, factors governing thymocyte motility, including cell speed (*Figure 1B*) and directionality (*Figure 1C*), were fine-tuned based on quantitative noninvasive imaging of the thymus using multiple medaka transgenic reporter lines (*Bajoghli et al., 2009*; *Bajoghli et al., 2015*; *Aghaallaei et al., 2021*). Based on these experimental observations, we modeled entry into the organ by creating new cells at the bottom of the simulated domain (ventral to the organ). These cells move into the thymus and, some time after differentiation, either leave the organ by exiting from the ventral side as a naïve T cell or undergo cell death by gradually shrinking and eventually being removed from the simulation (*Figure 1—videos 1–2*). The model accounts for the time taken for a single progenitor to enter and exit the thymus, which is estimated at approximately 3 days in zebrafish (*Hess and Boehm, 2012*) and medaka (*Bajoghli et al., 2015*). For time calibration, each simulation step was set to represent 15 s. With a total of 48,000 steps per simulation, the entire simulation spans 720,000 s, equivalent to about 8.3 days (*Figure 1D*). This time interval was chosen to recapitulate thymic development in medaka from 2.5 to 11 days post-fertilization (dpf), because at 2.5 dpf the first ETP enters the embryonic thymus (*Bajoghli et al., 2019*). Correspondingly, each simulation starts with entering the first ETP into the virtual thymus and ends with a homeostatic cell population of roughly 100 cells (mean ± standard deviation: 98±7; *Figure 1—video 1*). At 11 dpf, the medaka thymus contains 877±146 (*N*=7) thymocytes. Considering that the virtual thymus represents 1/10 of the medaka embryonic thymus, the cell population in the virtual thymus model reproduces the temporal dynamics in medaka within the margin of error.

Each cell's internal state and decision to proliferate or differentiate depends on intrinsic and extrinsic factors that are integrated by signaling pathways. The virtual thymus includes Delta-like 4 (Dll4), the ligand of Notch1 receptor, and interleukin-7 (IL-7) cytokine as two cell-extrinsic factors provided by the TECs (*Figure 1E*). Consistent with medaka WT embryonic thymus (*Aghaallaei et al., 2021*; *Aghaallaei et al., 2022*), all virtual TECs in this model express the Dll4 (*Figure 1E*, top-right panel), with a subset spatially releasing IL-7 into the environment (*Figure 1E*, top-left panel), creating a short-ranged IL-7 cytokine gradient in the thymic cortex niche (*Aghaallaei et al., 2021*). In terms of cell-intrinsic factors, all ETPs uniformly express the Notch1 receptor (*Figure 1E*, middle-right panel), and each cell lineage expresses the IL-7 receptor (IL7R) at a constant level chosen randomly between 0 and 1 (*Figure 1E*, middle-left panel). In each simulation, new ETPs continually enter the thymic niche at regular intervals. Following engagement with TECs and receiving Notch1 and IL-7 signals (*Figure 1E*, bottom panels), they proliferate and differentiate (step 3 in *Figure 1A*). The cell cycle was modeled with an average duration of 7 hr and was subdivided into G1, S, G2, and M phases. In line with in vivo observations (*Ruijtenberg and van den Heuvel, 2016*), virtual cells require external pro-proliferative signals during the G1 phase to commit to S through M. Once differentiated, cells can no longer enter the cell cycle. The dynamics of pro-proliferative signals and the proliferation stop after differentiation allow clones in silico to undergo up to 4 rounds of cell division (*Aghaallaei et al., 2021*). Thymic selection is modeled as a probabilistic event and is not regulated by signaling dynamics.

In this work, we implemented a code that enables us to meticulously trace the unique identity of each cell upon entering the virtual thymus and subsequently track the identity of its descendants (*Figure 1F*, left panel; *Figure 1—figure supplement 1C*). This tool identified various waves of clones

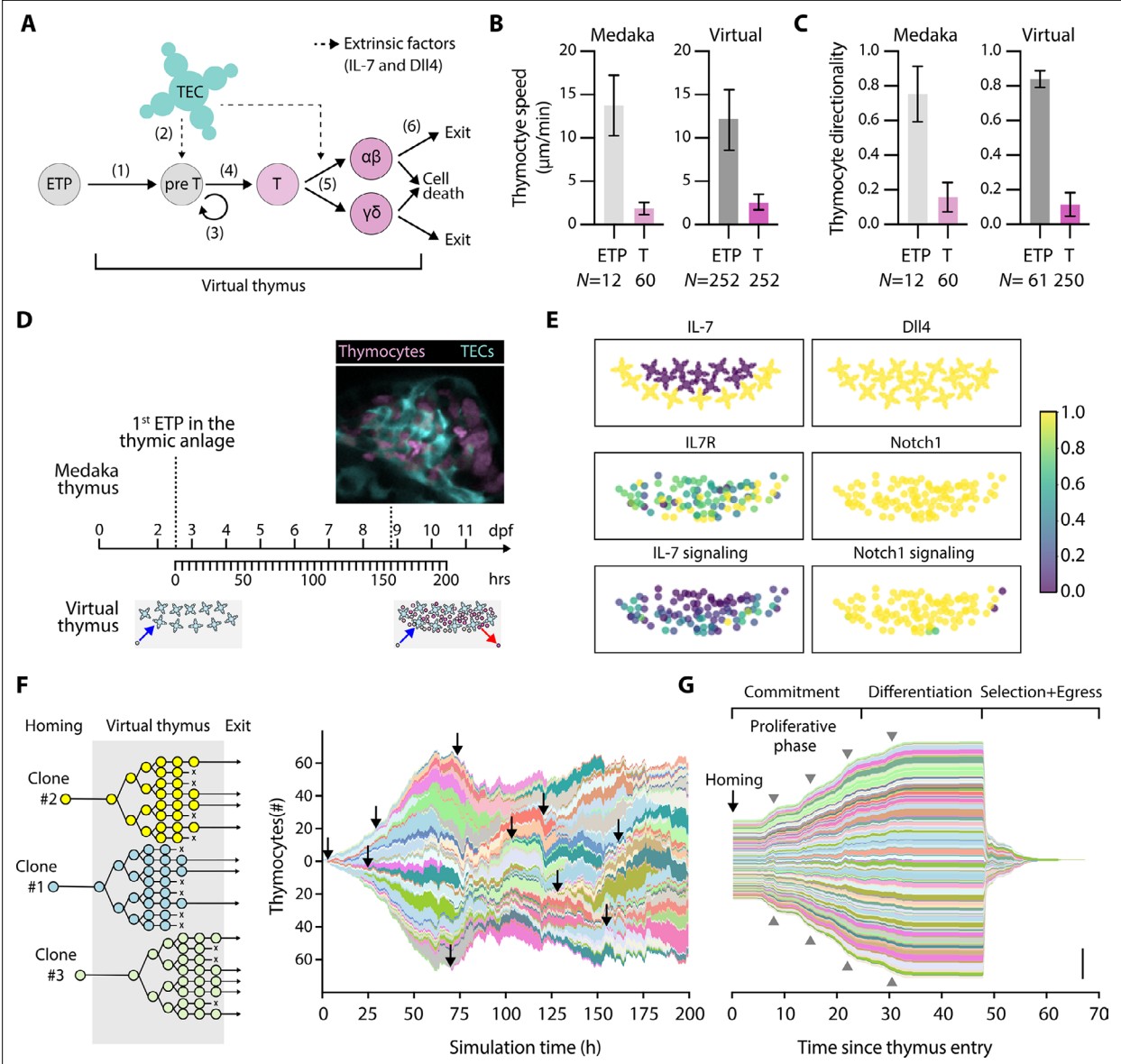

**Figure 1.** The virtual thymus model enables spatiotemporally resolved clonal analysis. (**A**) Schematic overview highlighting six key steps during T-cell development, which are integrated into the virtual thymus model: (1) ETPs enter the thymus niche; (2) they receive Dll4 and IL-7 signals from the thymic epithelial cells (TECs). Triggered by these signals, (3) thymocytes undergo proliferation and (4) differentiation into two distinct T-cell sublineages (5). Finally, thymocytes undergo selection, either leaving the organ or undergoing cell death (6). (**B–C**) Experimentally measured ('medaka') and computationally modeled ('virtual') cell speed (**B**) and cell directionality (**C**) for ETPs and thymocytes. Bars are mean values, and error bars standard deviation. (**D**) Timeline of medaka embryonic development (top) compared to simulated time (bottom). As indicated by the cartoon at the bottom, the first ETP enters the thymus roughly at 2.5 days post-fertilization (dpf) ($t$=0 hr in the simulation). At roughly 60 hr (roughly 5 dpf), the thymus reaches its homeostatic cell population, with both cell entry and exit. (**E**) Representative snapshots of the simulation at homeostasis showing spatial expression patterns and signaling activity of components of IL7R and Notch1 signaling pathways in TECs (top two panels) and thymocytes (bottom four panels). Data are shown in arbitrary units normalized to a scale from 0 to 1. (**F**) Left: Cartoon explaining the asynchronous arrival (homing) of new cells into the organ, their clonal expansion, and asynchronous cell death or exit. Right: Visualization of clonal diversity in a typical simulation; each color shade uniquely indicates a clonal lineage. The height of each colored patch indicates the number of cells in that clone, the length indicates simulation time. For illustrative purposes, a black arrow indicates the homing of a sample of lineages. (**G**) Data from the right panel in (**F**) was normalized to time of homing. This visualization highlights developmental phases of thymocyte lineages. Gray arrowheads indicate rounds of cell division. Scale bar: 50 cells. Abbreviations: ETP = early thymic progenitor, T=thymocyte, $N$=number of replicates (biological or computational).

The online version of this article includes the following video and figure supplement(s) for figure 1:

**Figure supplement 1.** The virtual thymus represents a slice of the organ.

*Figure 1 continued on next page*

*Figure 1 continued*

**Figure 1—video 1.** Simulation time-lapse.

https://elifesciences.org/articles/101137/figures#fig1video1

**Figure 1—video 2.** Simulation time-lapse with fewer cells to highlight clonal dynamics.

https://elifesciences.org/articles/101137/figures#fig1video2

---

because of the constant influx of new ETPs and negative selection and efflux of fully differentiated T cells (*Figure 1F*, arrows in the right panel, *Figure 1—video 1*). We note a slight variability in the clone sizes, which could arise from three processes. First, random variations in the expression level of IL7R impact IL-7 signal transduction and, consequently, proliferation (*Aghaallaei et al., 2021*). Second, due to random cell motility and cell crowding effects, certain cells could coincidentally remain in close contact with IL-7-secreting TECs, maximizing their exposure to pro-proliferative signals such as IL-7 and Notch ligand, and thus increasing their chances of entering the cell cycle. Third, each cell determines the duration of its next cell cycle by drawing an Erlang-distributed random variable with a mean of 7 hr and a standard deviation of $\cong$0.99 hr.

After normalizing all clones based on their time of entry into the thymus, the pattern of developmental phases became clear across all clones (*Figure 1G*). In this visualization, the now near-synchronous rounds of cell division across clones can be seen as bumps in the graph indicating clonal expansion (*Figure 1G*, gray arrowheads). During the commitment phase, which has a minimum duration of 24 hr but can be extended due to insufficient Notch signaling, thymocytes were competent for proliferation, leading to an increase in the total cell number per clone. In the differentiation phase, which has a fixed duration of 24 hr, clones reached a maximum size. Note that differentiation prevents further entry into the cell cycle but permits cells that passed G1 and therefore committed to S through M phases to complete their division. Finally, clones underwent selection and exit from the virtual thymus, thus leaving the simulation. Together, the enhanced virtual thymus model facilitated the investigation of the heterogeneity and temporal dynamics of individual clones originating from a founder progenitor cell.

## TEC architecture promotes thymocyte proliferation by modulating IL-7 availability

In vivo, individual TECs exhibit a distinctive star-shaped morphology, and protrusions of neighboring TECs contact each other forming a three-dimensional network (*Figure 2A*, left panel). Using confocal imaging, we estimated the relative area occupied by thymocytes (53 ± 3%; *N*=4) and TECs (47 ± 3%; *N*=4). A similar morphology and tissue density was replicated in our in silico model (*Figure 2A*, right panel), with thymocytes taking up on average 56±2% and TECs 44±2% of the volume in homeostasis. We wondered to what extent TEC morphology and density could impact thymocyte population size. This aspect is difficult to study experimentally and is, therefore, an ideal use case for the virtual thymus model. In a series of simulations, we combinatorially varied (1) the size of TECs, (2) the number of their protrusions, and (3) the TEC cell density (*Figure 2B*). The average homeostatic thymocyte population size in each simulation was then used as a readout. The results from 26 different tested conditions predicted that a higher density of larger TECs with more protrusions led to an almost twofold increase in the number of thymocytes (200±13, *N*=3; scenario 26 in *Figure 2C*, *Figure 2—figure supplement 1A*, bottom) compared to the reference condition (98±7, *N*=19; reference in *Figure 2C*). Conversely, scenarios where these parameters were reduced, particularly a reduced TEC size, also diminished the homeostatic thymocyte population size (scenarios 1–14 in *Figure 2C*). The condition with the least amount of cells, scenario 1, had a reduced TEC size, but an unchanged number of protrusions and density. The condition diametrically opposite to scenario 26 was scenario 6, with a reduced TEC size, reduced number of protrusions, and reduced TEC density (*Figure 2—figure supplement 1A*, top). In both scenario 1 (45±1, *N*=3) and scenario 6 (51±2, *N*=3), the thymic population size was only about half of that in the reference condition (98∓7, *N*=19). Statistical analysis confirmed that all three parameters and their interactions were significant predictors of total thymocyte population size (*Figure 2—figure supplement 1B and C*). The statistically significant interaction can be explained by the fact that higher TEC density increases the number of TECs and thereby amplifies the effect of an increased TEC

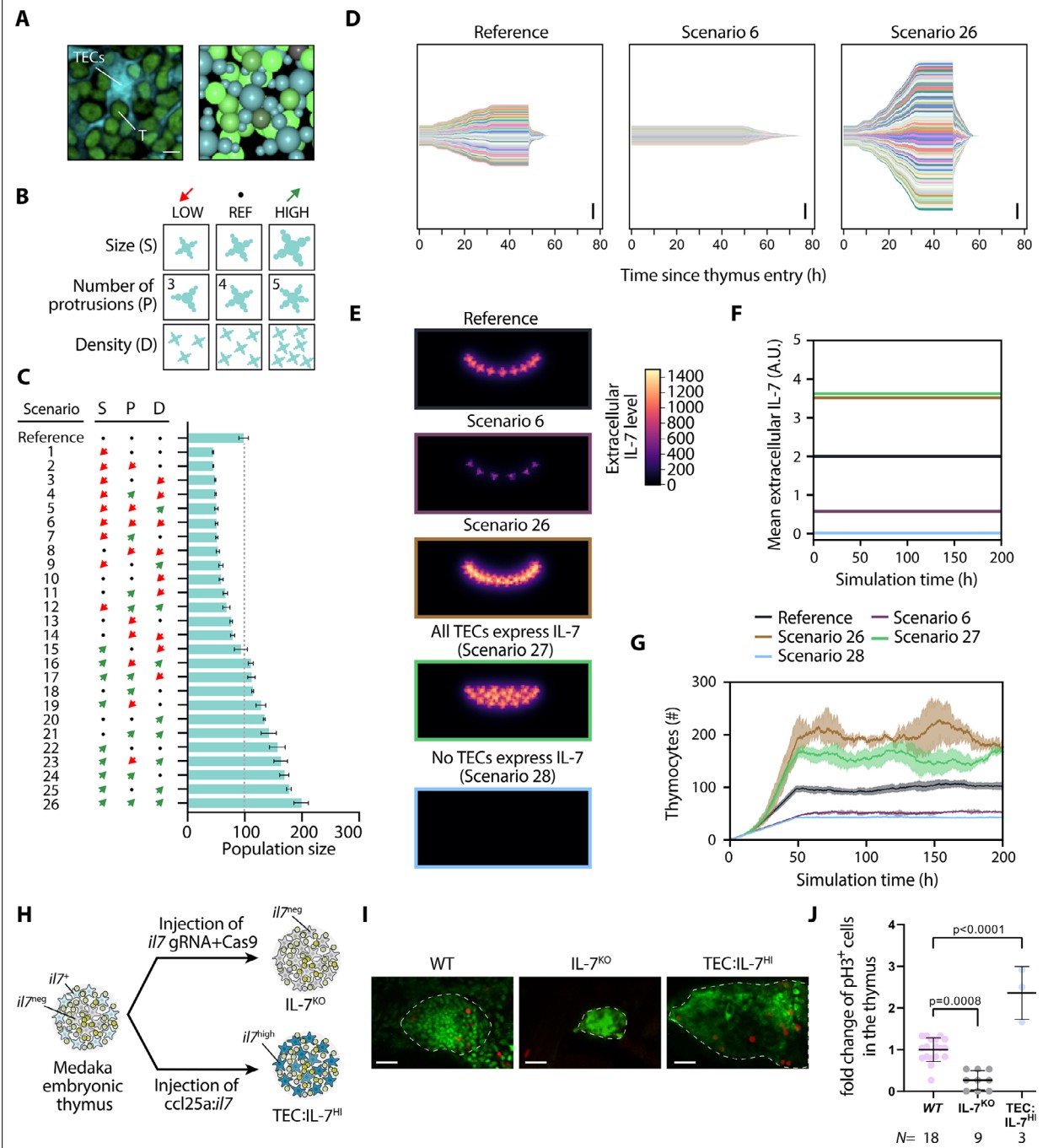

**Figure 2.** Thymic epithelial cell (TEC) density influences thymocyte proliferation via interleukin-7 receptor (IL7R) signaling. (**A**) Left: Detail of confocal section highlighting the tight spatial interaction between TECs and thymocytes (**T**). Scale bar: 5 µm. Right: Detail from a three-dimensional render of a simulation snapshot; TECs cyan, thymocytes green. (**B**) Parameter permutations tested. (**C**) Impact of parameter permutations on average homeostatic cell population. Bars indicate means, error bars standard deviation. The abbreviations 'S', 'P', and 'D' correspond to parameters 'Size', 'Number of Protrusions', and 'Density', respectively, as in panel (**B**). (**D**) Clonal lineages normalized to time of thymic entry for reference (left), a scenario with low average population (middle) and a scenario with high average population (right). Scale bar: 100 cells. (**E**) Extracellular IL-7 gradient in simulations; sum projection of all z-planes of a simulated confocal stack. Units are arbitrary. (**F**) Mean extracellular IL-7 concentration in the entire simulated volume. Note that these values are lower than in (**E**) because of averaging over the volume, including empty space (black areas in (**E**)). (**G**) Thymocyte population size over time averaged over several simulation runs. Shaded area indicates standard deviation. (**H**) Illustration of experimental setup for IL-7[KO] and TEC:IL-7[HI]. At least three technical replicates were done for the injection of each construct. (**I**) Representative images of embryos stained with green fluorescent protein (GFP) (green) and phospho-histone 3 (pH3) (red) of wild-type (WT), IL-7[KO], and TEC:IL-7[HI]. Scale bar: 20 µm. (**J**) Numbers of pH3 positive cells per thymus normalized to mean of WT. *N* represents the number of individual fish. Data show means ± standard deviation.

*Figure 2 continued on next page*

*Figure 2 continued*

The online version of this article includes the following figure supplement(s) for figure 2:

**Figure supplement 1.** Statistical analysis of simulated data.

**Figure supplement 2.** Reduced thymus volume upon IL-7$^{KO}$.

radius or number of protrusions. Thus, these three alterations of the thymic niche morphology act synergistically to modulate thymocyte population size.

There are two possibilities that could explain the variations in total thymocyte numbers in simulations: either over- or under-proliferation of a few thymocyte lineages or the cumulative impact of subtle changes across several thymocyte clones. To distinguish between these possibilities, we used our clonal analysis tool for scenarios 6 and 26 (*Figure 2D*). This analysis revealed that variations in total thymocyte numbers were a consequence of changes in proliferation across all clones to a similar degree. These results predict that the TEC architecture has a direct impact on thymocyte proliferation rate, which fits well with the fact that TECs act as the main source of ligands, growth factors, cytokines, and chemokines (*Gameiro et al., 2010*). Any changes in the TEC architecture could therefore influence the amount of cytokine production. In our virtual thymus model, this affects the extracellular spatial distribution of IL-7 cytokine (*Figure 2E*). To further explore this, we tested two additional scenarios. In scenario 27, TECs maintained their reference architecture and density but all expressed IL-7, leading to a more uniform cytokine distribution in the thymic environment (*Figure 2E*). The extracellular level of IL-7 (*Figure 2F*) and thymocyte population size (155±11, *N*=5; *Figure 2G*) in scenario 27 was similarly high as in scenario 26, where the size, protrusion number, and density of TECs were increased. Conversely, when TECs did not express IL-7 (scenario 28), the thymocyte population size (*Figure 2G*) was low (42±1, *N*=5), akin to scenario 6, where the size, protrusion, and density of TECs were reduced. Interestingly, although the average extracellular IL-7 level is at its highest in scenario 27 with ubiquitous expression, the spatially restricted elevated IL-7 concentration in scenario 26 is more effective at driving thymic population growth (*Figure 2E–G*). Thymocytes first enter the organ at its periphery from below (at the ventrolateral site), where IL-7 levels are highest, and tend to migrate to the IL-7-depleted center of the thymus as they become non-proliferative (*Aghaallaei et al., 2021*; see also *Figure 1—video 1*). Thus, in scenario 26, thymocytes in their proliferative phase immediately encounter elevated IL-7 at the organ periphery, stimulating cell proliferation. In contrast, in scenario 27, thymocytes that enter the organ periphery encounter comparable levels of IL-7 to the reference scenario, and only after entering deeper into the organ can the ubiquitous expression of IL-7 unfold its effects. This delay, contingent to random cell migration (which is further exacerbated by cell crowding posing an obstacle), reduces the impact of elevated IL-7 in scenario 27. These results highlight how the spatial structure of the thymic niche can impact thymocyte population dynamics.

Our simulations predict that the availability of IL-7 provided by TECs has a direct impact on thymocyte population size. To validate this prediction in vivo, we conducted two functional analyses using the medaka model organism (*Figure 2H*). In one experiment, employing the CRISPR-Cas9 technique, we knocked out the *il7* gene (*Figure 2—figure supplement 2A and B*) in a medaka transgenic line, where thymocytes express green fluorescent protein (GFP). Consistent with our in silico outcomes, *il7* crispant embryos displayed fewer thymocytes and a smaller thymus size (*Figure 2I*, *Figure 2—figure supplement 2C and D*). To estimate the extent of cell proliferation, we counted mitotic cells throughout the entire thymus using the M phase marker phospho-histone 3 (pH3) and normalized values to the mean of the control. This analysis further supported that a reduction in IL-7 availability in the thymus reduces thymocyte proliferation (*Figure 2I and J*). In a second experimental setup (*Figure 2H*), we artificially increased *il7* levels in the thymic niche by injecting the ccl25a:*il7* construct into embryos (hereafter called TEC:IL-7$^{HI}$), resulting in a strong upregulation of the levels of *il7* produced by the TECs (*Aghaallaei et al., 2021*). Compared to WT, the thymus of TEC:IL-7$^{HI}$ embryos appeared visibly larger, and the number of pH3 positive cells was significantly increased (*Figure 2I and J*). Therefore, the in vivo results confirm the critical role of IL-7 for thymocyte proliferation in medaka, a mechanism that is evolutionarily conserved among vertebrates (*Iwanami et al., 2011*). Additionally, the alterations in thymic size suggest a regulatory crosstalk between thymocyte proliferation and TEC architecture.

## Cell-autonomous sensitivity to IL-7 signaling promotes thymocyte proliferation

Given the important contribution of extracellular IL-7 to thymocyte population size, we next evaluated the impact of (i) extracellular IL-7 depletion (e.g. via ligand internalization), (2) the rate of signal transduction activation upon IL7R and IL-7 binding, and (3) the rate of decay of signaling activity (*Figure 3A and B*). In our virtual thymus model, the IL-7 signaling activity ($\sigma_{IL7}$) of a thymocyte over time (*t*) is modeled phenomenologically using a function involving the IL7R concentration ([IL7R]), the average extracellular IL-7 concentration ( $\langle[IL\text{-}7_{ex}]\rangle$ ), a parameter ($a_{IL\text{-}7}$) that scales the strength of signal transduction activation, and another parameter ($d_{IL\text{-}7}$) that scales the rate at which the signaling activity diminishes over time (*Figure 3B*). As before, we combinatorially perturbed these three factors, simulating 17 different scenarios, to assay their impact on thymocyte population size (*Figure 3C*). As expected, we observed a positive correlation between the level of signal transduction activation $a_{IL\text{-}7}$ and thymocyte numbers. Conversely, variations in signaling decay rate $d_{IL\text{-}7}$ showed the opposite effect. The rate of signal transduction deactivation models cellular short-term memory, thus a lower signaling deactivation rate indicates that cells retain their IL-7 stimulus for a longer duration. Because IL-7 signaling activity $\sigma_{IL7}$ promotes cell cycle entry (see Appendix 1 section *X. Subcellular scale: cell proliferation model* for an in-depth explanation), we expect that parameter changes that increase stimulus duration should promote cell division. Indeed, the scenario with the highest number of thymocytes occurred when signaling deactivation was low and activation was high (274±11, *N*=4; scenario 45 in *Figure 3C*). The diametrically opposite combination had the least amount of cells (42±1, *N*=4; scenario 29 in *Figure 3C*). Statistical analysis confirms that signal transduction activation $a_{IL\text{-}7}$ and the signal transduction deactivation rate $d_{IL\text{-}7}$ were both statistically significant predictors of thymocyte population size (*Figure 3—figure supplement 1*). There was no statistically significant effect of parameter interactions, likely because these parameters exert their effect independently. There was only a limited reduction in population size with IL-7 depletion (91±11, *N*=4; scenario 36 in *Figure 3C*) compared to no depletion in our reference setting (98±7, *N*=19). Overall, IL-7 depletion did not have a significant effect on thymocyte population (*Figure 3—figure supplement 1*). However, we noted that population size difference was more pronounced at higher cell numbers, e.g., comparing between scenario 44 (237±7, *N*=4) and scenario 45 (274±11, *N*=4). Similarly, the mean extracellular IL-7 content in the virtual thymus barely changed when comparing the reference to scenario 36 but was noticeably reduced in scenario 44 compared to scenario 45 (*Figure 3D*). Therefore, thymocyte proliferation could self-inhibit via depletion of extracellular IL-7, but we expect this effect to be small unless cell numbers are massively increased.

Together, the outcomes of our simulations reveal that molecular and cellular changes, specifically those capable of increasing the extrinsic factor IL-7 in the niche – such as a higher density of TECs or an elevated expression level of IL-7 by individual TECs – directly contribute to an increase in thymocyte population size. Tuning thymocytes' sensitivity to IL-7 by manipulating IL7R signaling can further amplify proliferation. Finally, we expect that cytokine internalization by thymocytes has only a mild inhibitory effect on excessive population growth by limiting the availability of the pro-proliferative IL-7 in the extracellular space.

## A systematic approach identifies TEC architecture as a synergistic factor promoting clonal expansion of IL7R-lesioned clones

Our 45 tested scenarios thus far reached a maximum threefold induction of thymocyte population size when parameter changes promoted increased IL-7 signaling. While the activation of IL-7 signaling is linked to T-ALL development, this pathology is associated with proliferation of clones carrying somatic mutations (*Oliveira et al., 2019*; *Aghaallaei et al., 2022*; *Silva et al., 2021*). We therefore modified the code of our virtual thymus model to enable the introduction of parameter changes exclusively in a single clone and all of its progeny (hereafter referred to as the lesioned clone). Inspired by observations that 10% of T-ALL patients exhibit dominant active mutations in the *IL7R* gene (*Liu et al., 2017*; *Shochat et al., 2011*; *Zenatti et al., 2011*), and a significant subset of T-ALL patients with active IL7R signaling display elevated *IL7R* expression levels (*Silva et al., 2021*), two types of lesions were introduced: (1) a dominant active IL7R lesion (hereafter called IL7R[DA] lesion), (2) an IL7R overexpression lesion (hereafter called IL7R[HI] lesion). For the IL7R[DA] lesion, we set the IL7R level of the lesioned clone to the maximum WT level of 1 and allowed it to activate IL-7 signaling regardless of extracellular IL-7

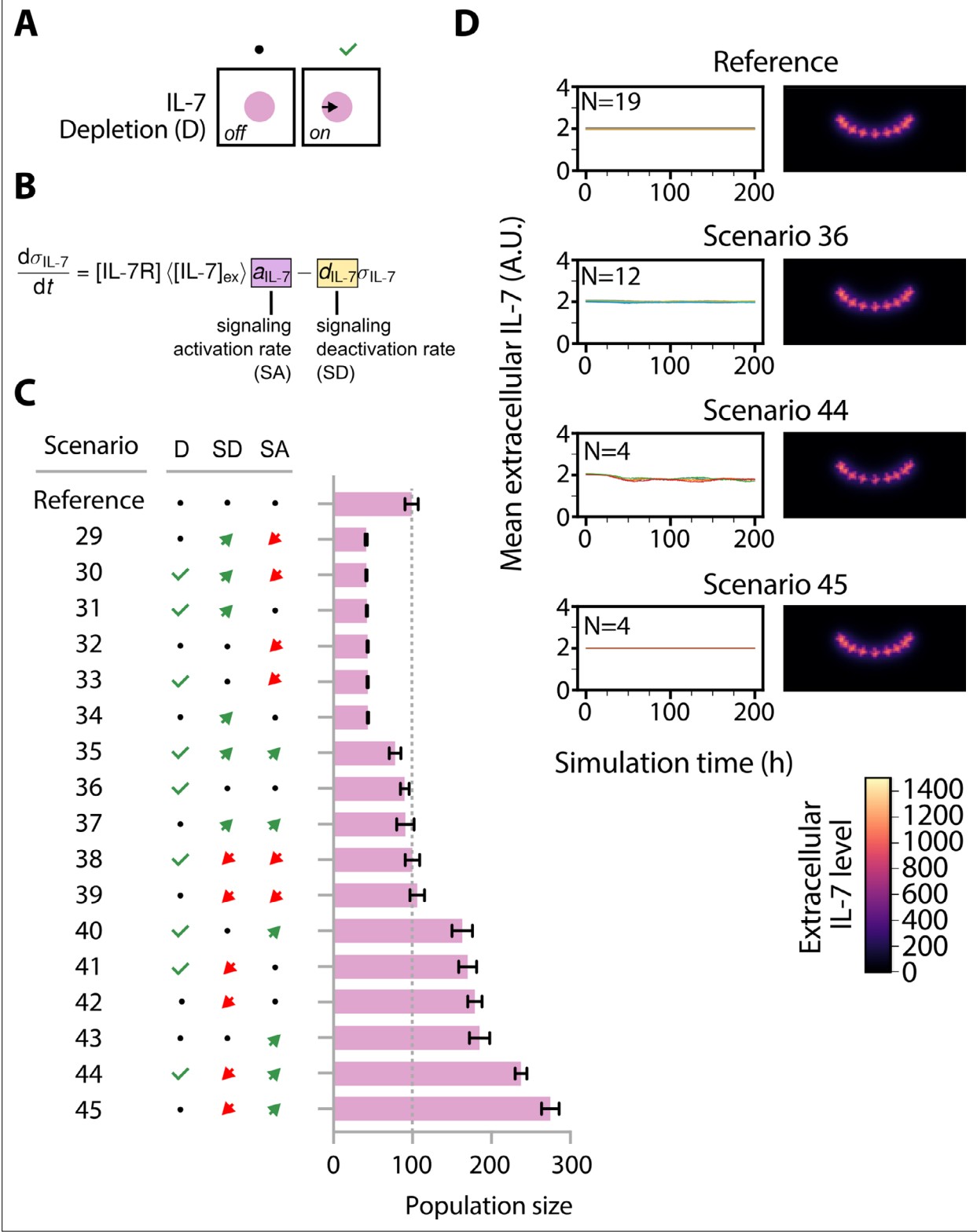

**Figure 3.** High interleukin-7 receptor (IL7R) signaling promotes proliferation. (**A**) Illustration of scenarios without extracellular IL-7 depletion (left) and with extracellular IL-7 depletion (right). (**B**) Ordinary differential equation used to model the IL7R signaling activity. Parameters that were modified are highlighted. (**C**) Effect of IL7R signaling-related parameter permutation. Bars indicate means and error bars standard deviation. The abbreviations 'D', 'SD', and 'SA' correspond to 'Depletion', 'Signaling Deactivation Rate', and 'Signaling Activation Rate', respectively, as indicated in panels (**A**) and (**B**). (**D**) Left: Mean extracellular IL-7 concentration in the entire simulated volume. Right: Extracellular IL-7 gradient in simulations; sum projection of all

*Figure 3 continued on next page*

*Figure 3 continued*

z-planes of a simulated confocal stack. Units are arbitrary. The introduction of IL-7 depletion by thymocytes leads to a slight reduction in extracellular IL-7 availability.

The online version of this article includes the following figure supplement(s) for figure 3:

**Figure supplement 1.** Statistical analysis of simulated data.

cytokine, mimicking the dominant activation mutations in the *IL7R* gene (*Figure 4A*, middle panel). For the IL7R$^{HI}$ lesion, we mimicked overexpression of the *IL7R* gene by assigning a value of 10 for the IL7R level in the lesioned clone, i.e., 10-fold higher than the maximum attainable in the WT population (*Figure 4A*, right panel). In both scenarios, non-lesioned thymocytes retained reference IL7R levels randomly ranging between 0 and 1 (*Aghaallaei et al., 2021*). In each simulation, a single lesioned clone entered the thymic niche after the 60th hour of a simulation, i.e., shortly after the establishment of a homeostatic population size (*Figure 4B*, *Figure 4—video 1*).

First, we evaluated the impact of IL7R$^{HI}$ and IL7R$^{DA}$ lesions in the reference condition, meaning no additional modifications were made to our model. The clonal analysis of simulations revealed that introducing a lesioned clone did not markedly alter the division behavior of non-lesioned clones in the same simulation (scenario with no lesion: 1.04±1.26 rounds of cell division; scenario with IL7R$^{DA}$: 0.98±1.22 rounds of cell division; scenario with IL7R$^{HI}$: 1.04±1.23 rounds of cell division; *Figure 4B and C*). The large variance in non-lesioned clones is consistent with our previous work (*Aghaallaei et al., 2021*), which showed that clones expressing endogenously high levels of IL7R undergo more rounds of cell division in the virtual thymus, whereas clones with lower levels of IL7R proliferate very little or not at all. In contrast, lesioned clones averaged just above 4 rounds of cell division (4.20±0.34 for IL7R$^{DA}$ and 4.10±0.13 for IL7R$^{HI}$). Consequently, the size of IL7R$^{DA}$ and IL7R$^{HI}$ clones was nearly eight-fold larger than an average non-lesioned clone (*Figure 4C*). This result suggests that lesioned clones acquired a distinct proliferative advantage over non-lesioned clones in our virtual thymus model. Nevertheless, since the expansion of lesioned clones amounted to, at best, only one additional round of cell division compared to the upper range of the non-lesioned clone distribution (*Figure 4C*), our virtual thymus model predicts that a single lesion in the *IL7R* gene will not be clinically impactful.

Therefore, we next explored whether additional modifications in both cell-autonomous and non-autonomous factors might substantially enhance the clonal expansion. To identify these factors, we decided to undertake a systematic approach and tested combinations of scenarios affecting the TEC architecture and IL-7 signaling parameters, as tested in *Figures 2 and 3*, together with changing other parameters affecting proliferation and differentiation of all clones (*Figure 4D*). Furthermore, we considered scenarios in which only lesioned clones exhibited additional modifications, such as reduced cell motility, slower differentiation time, or autocrine IL-7 production (*Table 1*). The inclusion of the latter was inspired by a study showing that certain malignant T cells derived from T-ALL patients possess the ability to ectopically express IL-7 cytokine (*Buffière et al., 2019*). In theory, there are 104,976 possible permutations of these scenarios. To reduce the space of possibilities and thus the computational cost, we prioritized alterations expected to increase proliferative potential based on outcomes shown in *Figures 2 and 3*. We also decided to focus on simulating the extremes; e.g., low and high levels shown in *Figure 4D*. In addition, we simulated a small sample of scenarios expected to decrease proliferative potential, such as a reduction in TEC density. In the end, 1580 scenarios were simulated. The clonal analysis tool was used to compare clone size and the number of cell division rounds between lesioned and non-lesioned clones under the same conditions. Overall, we observed a similar proliferation pattern among IL7R$^{HI}$ and IL7R$^{DA}$ clones in most of the tested scenarios. Likewise, non-lesioned clones displayed similar trends as clones in control simulations that lacked a lesioned clone (*Figure 4—figure supplement 1*). The lesioned clones consistently showed a higher proliferation rate than their non-lesioned counterpart (*Figure 4E*; *Figure 4—source data 1*). This trend persisted even when parameters known to generally increase cell division for all cells, such as mean cycle duration and proliferation duration, were modified (*Figure 4—figure supplement 2*, *Figure 4—video 2*). In fact, lesioned clones disproportionally gained up to 6 rounds of cell division by modulating proliferative parameters, while normal clones only gained 2 rounds of cell division in the same scenarios. This difference results from cell crowding creating a disproportional disadvantage to non-lesioned clones: In the scenario with increased proliferation, the first clones to colonize the organ can massively proliferate and completely surround the TECs, preventing later arriving clones

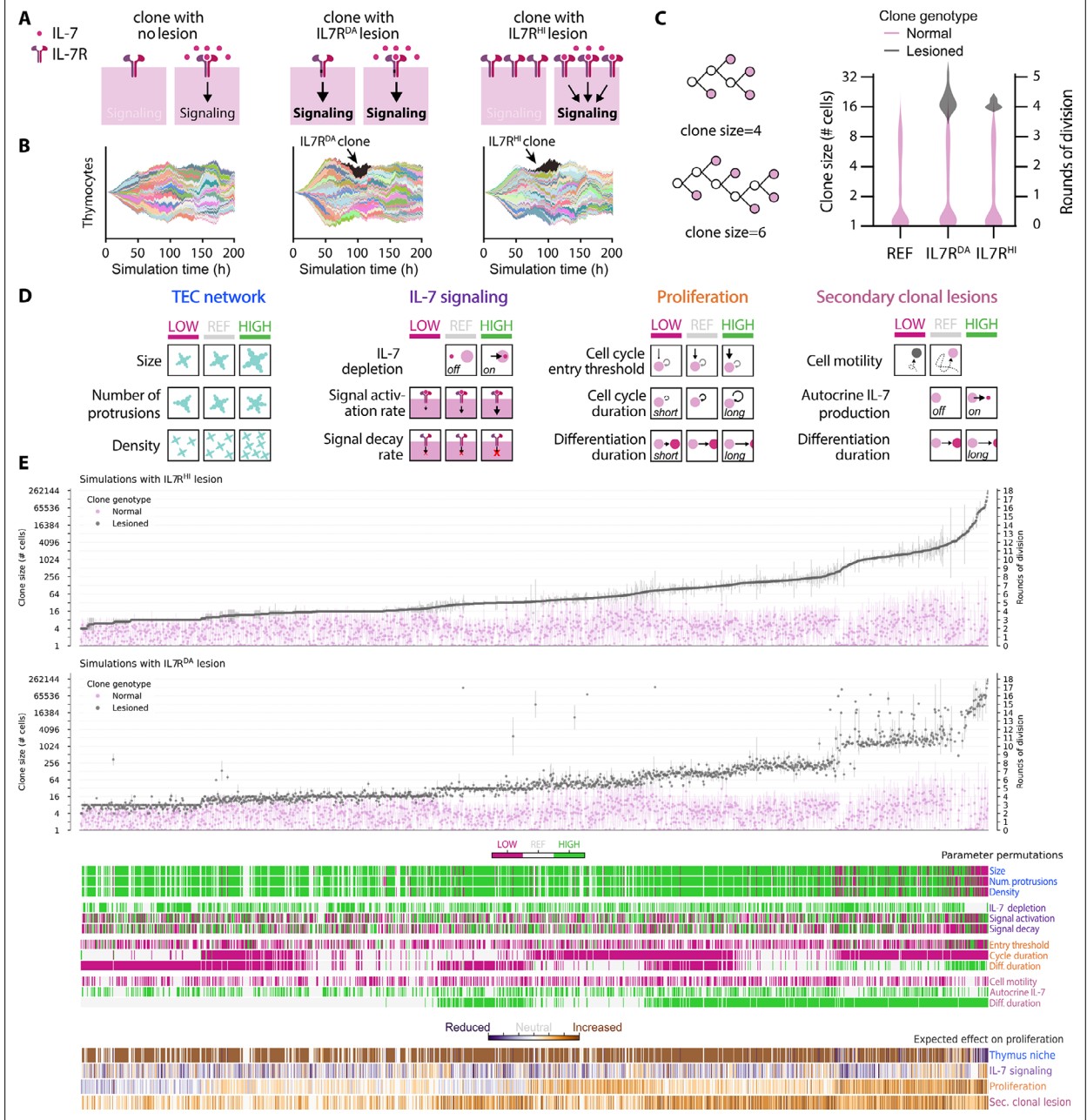

**Figure 4.** Systematic screen shows that lower thymic epithelial cell (TEC) density promotes malignant expansion. (**A**) Schematic representation of the major lesions to interleukin-7 receptor (IL7R) that were implemented in the model. (**B**) Representative clonal plot highlighting in black the lesioned clone. (**C**) We defined clone size as the maximum number of terminal leaves of a lineage, regardless of eventual fate of each cell. Clone size distribution in the reference simulation (REF), compared to IL7R^DA and IL7R^HI. REF: *N*=20 simulations, with 2088 clones total: IL7R^DA: *N*=13 simulations with 1330 non-lesioned clones and 13 lesioned clones. IL7R^HI: *N*=13 simulations with 1336 non-lesioned clones and 13 lesioned clones. (**D**) Schematic of tested scenarios shown in (**E**). (**E**) Simulated permutations ordered along increasing mean lesioned clone size for the IL7R^HI condition. The expected effect on proliferation is a qualitative estimate based on preliminary simulations. A total of 1580 permutations were tested with a varying amount of replicates per permutation. Error bars indicate standard deviation. Note that in (**C**) and (**E**) the axis is in base-2 logarithmic scale to better illustrate rounds of cell division.

The online version of this article includes the following video, source data, and figure supplement(s) for figure 4:

**Source data 1.** Source data file for visualization in *Figure 4E* and *Figure 4—figure supplement 1B*.

**Figure supplement 1.** Clones in control simulations show similar sensitivity to parameter alterations as non-lesioned clones.

**Figure supplement 2.** Modulation of proliferation-related parameters affects lesioned thymocytes more strongly.

*Figure 4 continued on next page*

*Figure 4 continued*

**Figure supplement 3.** Modulation of parameters affecting IL7-signaling has almost no effect on lesioned clones.

**Figure supplement 4.** A secondary lesion delaying differentiation doubles proliferative potential.

**Figure supplement 5.** Parameters affecting thymic epithelial cell (TEC) distribution have opposing effects on lesioned and non-lesioned clones.

**Figure supplement 6.** Sparse thymic epithelial cell (TEC) distribution is among permutations that most enabled lesioned clone expansion.

**Figure 4—video 1.** Simulation time-lapse with lesioned clone.

https://elifesciences.org/articles/101137/figures#fig4video1

**Figure 4—video 2.** Simulation time-lapse.

https://elifesciences.org/articles/101137/figures#fig4video2

from accessing TEC-derived pro-proliferative signals (*Figure 4—video 2*, right column). Thus, these few 'lucky' early colonizers prevent late cells from attaining their full proliferative potential. Lesioned clones carry an intrinsic pro-proliferative advantage due to their lesions in the IL7R, enabling proliferation despite a lack of direct contact with TECs. In contrast, modifying parameters that affected IL-7 activity only enhanced the proliferation rate of non-lesioned clones but showed no impact on either IL7R$^{HI}$ or IL7R$^{DA}$ clones (*Figure 4—figure supplement 3*). While this outcome is expected for IL7R$^{DA}$ clones, which are insensitive to IL-7, for IL7R$^{HI}$ clones, it indicates that 10-fold receptor overexpression is enough to activate downstream pro-proliferative effects of the pathway regardless of the parameter permutations that we tested. In contrast, proliferation of normal clones is strongly affected in these conditions, gaining up to 3 rounds of cell division between least and most proliferative conditions.

In our systematic approach, we did not detect an added effect on clonal expansion from combining reduced cell motility or autocrine IL-7 production with IL7R$^{HI}$ or IL7R$^{DA}$ lesions (*Figure 4—figure supplement 4*). Added IL-7 secretion by lesioned clones with the autocrine lesion had only a small effect on proliferation of non-lesioned clones in the same simulation. The most critical lesion in conferring a proliferative advantage to lesioned clones was a delay in differentiation. This delay essentially doubled the duration of the proliferative phase, and indeed we observed up to 8 rounds of cell division, double the level of clones having only a single lesion in the IL7R. This result fits well with in vivo data, suggesting that perturbations in differentiation have been implicated in contributing to the initiation and progression of cancer (*Ruijtenberg and van den Heuvel, 2016*).

Surprisingly, the expansion of lesioned clones was amplified when the TEC niche was at its sparsest – a result opposite to the non-lesioned clones which instead profit from a dense TEC niche (*Figure 4—figure supplement 5*). While non-lesioned clones gained 2 rounds of division with an increase in TEC density, lesioned clones lost 2 rounds of division. Indeed, parameter combinations with reduced density and size of TECs were overrepresented in the scenarios that most increased proliferation (*Figure 4—figure supplement 6A–C*). For example, in the most extreme scenario observed in our systematic approach, IL7R$^{HI}$ or IL7R$^{DA}$ clones underwent up to 18 rounds of cell division – far above the 12 rounds predicted from modifying each group of parameters in isolation (+6 from proliferation, +4 from added lesions, +2 from reduced TEC density, assuming an additive effect; *Figure 4—figure supplements*

**Table 1.** Clonal lesions and their effects.

| Lesion | Explanation |
|---|---|
| IL7R$^{WT}$ | The concentration of IL7R receptor in the lesioned clone was set to the maximum attainable in wild-type thymocytes. Thus, this lesion is representative of an extreme example among the non-lesioned population and was included as a control. |
| IL7R$^{DA}$ | This concentration of IL7R receptor in the lesioned clone was set to the maximum attainable in wild-type thymocytes. Moreover, the receptor was constitutively active, irrespective of the presence of IL-7 ligand. |
| IL7R$^{HI}$ | The concentration of IL7R receptor in the lesioned clone was set to 10-fold higher than the maximum attainable in wild-type thymocytes. |
| IL-7 autocrine | Lesioned clones secrete IL-7 at the same rate as TECs, acting as additional sources in *Equation 4*. |
| NOTCH1$^{DA}$ | Lesioned clones had constitutively active Notch signaling irrespective of the presence of Delta ligand. The levels of Notch signaling were set to the maximum attainable in non-lesioned thymocytes and could not be reduced. |
| Delayed differentiation | The lesioned clone had its rate of differentiation halved, effectively doubling the duration of the proliferative phase. |
| Slower cell speed | The lesioned clone's maximum speed was half of the non-lesioned population's maximum speed. |

*2–5*). This result suggests that a defective niche amplifies the effect of other pro-proliferative modulations such as slower differentiation and shorter cell cycle, conferring a synergistic advantage to IL7R[HI] and IL7R[DA] clones compared to their normal counterparts. Note that at 18 rounds of division, lesioned clones were so massive that the total lesioned cell volume was over a 1000-fold larger than the volume of the simulated organ slice (*Figure 4—figure supplement 6D*). In comparison, in the homeostatic condition, the total volume occupied by both thymocytes and TECs amounted to only 0.7-fold of the available tissue volume. Though our simulations by default include physical volume exclusion effects that prevent dense packing, we did not implement density-dependent feedback on proliferation, enabling cells to duplicate their volume at cell division and proliferate uncontrolled despite the lack of available space. Presumably, similar conditions in vivo would lead to thymus hyperplasia. In support of this view, our experimental manipulations of IL-7 levels in the thymus had a direct effect on organ size (*Figure 2I*, *Figure 2—figure supplement 2C and D*).

Identifying a sparse TEC network as a new factor that could influence the clonal expansion of lesioned clones was an unexpected outcome. This is because, in the WT situation, a sparser TEC network results in decreased extracellular IL-7, leading to reduced clonal size (*Figure 2*). While being disadvantageous for non-lesioned clones, this compromised thymic niche proved to be optimal for a massive expansion of lesioned clones. Conversely, we observed that a denser TEC network – characterized by a higher density of TECs with more protrusions and larger size – positively influenced the population size of non-lesioned cells but had a negative impact on the IL7R[HI] and IL7R[DA] lesioned clones (*Figure 4—figure supplement 5*). Similarly, in conditions of high cell crowding, lack of contact with TECs put non-lesioned clones at a disadvantage, while lesioned clones appeared to benefit (*Figure 4—figure supplement 2*, *Figure 4—video 2*). This outcome might be explained by the signaling dynamics within our model implementation, where the engagement of the NOTCH1 receptor on thymocytes with the DLL4 ligand on TECs, on the one hand, promotes proliferation and, on the other hand, is a prerequisite for the differentiation process until commitment to a nondividing fate. In the model, if the NOTCH1 receptor fails to engage with DLL4 on TECs due to their low density or cell crowding effects, cells will remain in an undifferentiated state for a longer time. A closer inspection of time-normalized clones in scenario 6 (very sparse TECs) and in scenario 26 (very dense TECs) indeed confirms that the thymocyte population has a longer turnover time when TECs are sparse (almost 80 hr, *Figure 2D*, middle panel) and a shorter turnover when TECs are dense (less than 60 hr, *Figure 2D*, right panel), indicative of the effect of Notch signaling. Together, cells in an environment with fewer TECs and thus lower Notch signaling will experience a prolonged proliferative phase but still require pro-proliferative signals to commit to the cell cycle. Owing to the higher levels of IL7R signal intrinsic to the lesion, lesioned clones will be more easily competent to proliferate even without Notch signaling and, therefore, benefit from a lower TEC density.

## The interplay between IL7R and NOTCH1 signals in the clonal expansion of thymocytes

In our virtual thymus model, ETPs simultaneously assess whether the combined sum of IL7R and NOTCH1 signals surpasses a threshold required for entry into the cell cycle (*Aghaallaei et al., 2021*; *Figure 5A*). Besides its effect on proliferation, NOTCH1 signaling also promotes ETP differentiation (*Aghaallaei et al., 2021*; *Aghaallaei et al., 2022*), and we used this fact and the observations in *notch1b* medaka mutant phenotypes to include an accelerating effect of NOTCH1 on differentiation in our model (*Aghaallaei et al., 2021*; see also Appendix 1). Thus, NOTCH1 has a dual effect in thymocytes, which we implemented in our model as an incoherent feed-forward loop (*Figure 5A*): NOTCH1 signaling promotes proliferation and accelerates differentiation, while differentiation inhibits proliferation by preventing further entry into the cell cycle. IL7R signaling promotes proliferation independently of NOTCH1 and has no direct or indirect effect on the differentiation process. To further explore the interplay between these two factors, we compared scenarios where lesioned clones exhibited only constitutive activation of the NOTCH1 receptor (hereafter called NOTCH1[DA]) or in combination with either IL7R[DA] or IL7R[HI]. The simulations predicted that lesioned clones with only NOTCH1[DA] modification displayed a very small advantage compared to non-lesioned clones; this increase in proliferation was only marginally higher than expected for a clone expressing the highest endogenous levels of IL7R (hereafter IL7R[WT]; *Figure 5B*). In contrast, lesions that specifically delay differentiation led to higher proliferation. Indeed, the addition of NOTCH1[DA] modification to IL7R[DA]

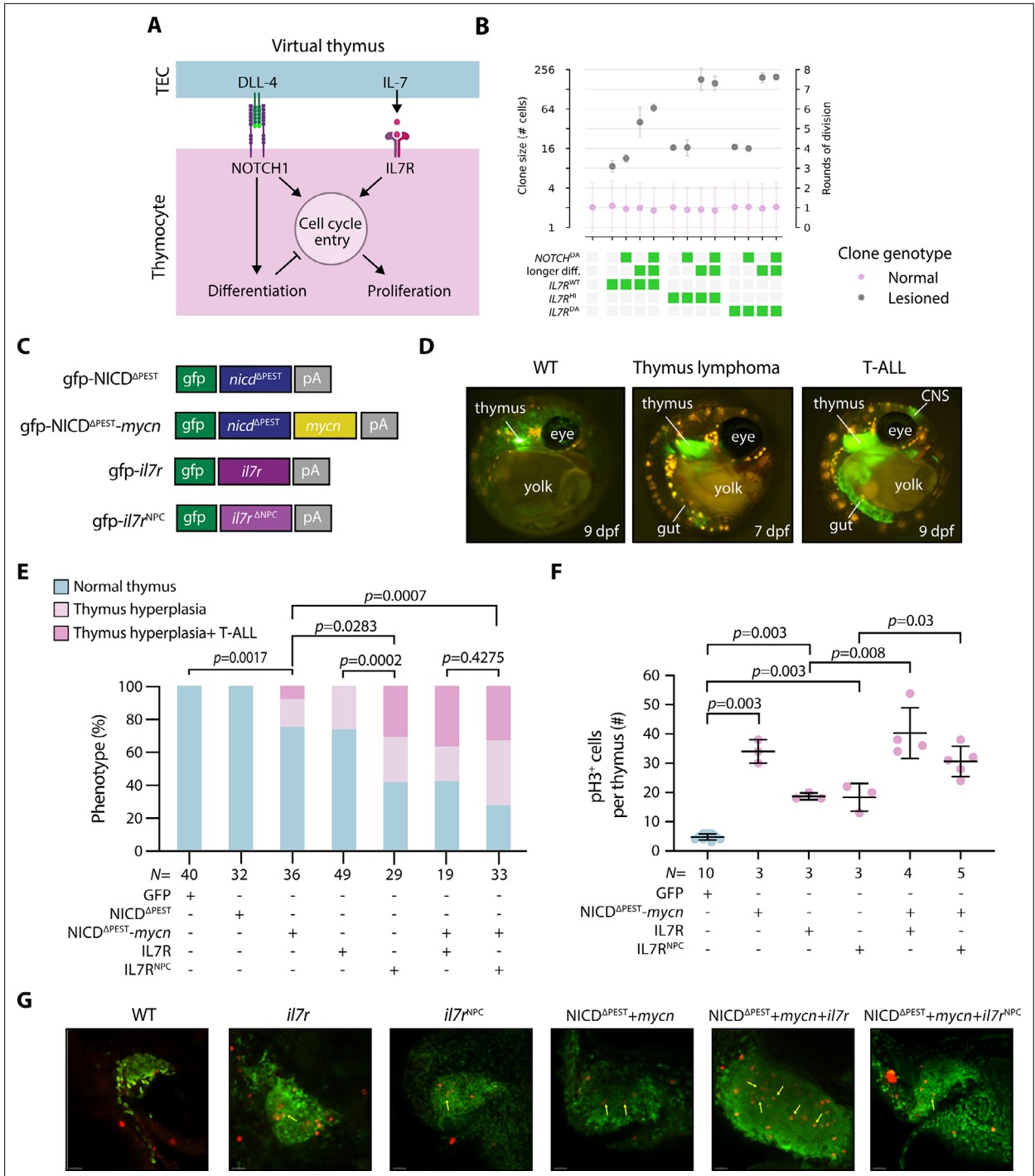

**Figure 5.** Interplay of interleukin-7 receptor (IL7R) and NOTCH1 potentiates clonal expansion in vivo. (**A**) Schematic of Notch1 and IL7R signaling and their regulation of thymocyte cell cycle entry, proliferation, and differentiation as implemented in the virtual thymus model. (**B**) Simulated data; rounds of cell division of normal and lesioned clones with different types of lesions in NOTCH1 or IL7R signaling. Note that the y-axis is base-2 logarithmic. Green boxes: condition/lesion present. Gray boxes: condition/lesion absent. (**C**) Illustration of plasmids for DNA microinjection for transient overexpression of the different genes. Note that the expression of green fluorescent protein (GFP) was used to select successfully injected embryos. (**D**) Representative image: (left) of the WT control at 9 days post-fertilization (dpf) where thymocytes are labeled in green with normal thymus size; (middle) of a thymus hyperplasia phenotype at 7 dpf; (right) same embryo developed T-cell acute lymphoblastic leukemia (T-ALL) phenotype at 9 dpf, where infiltration in other organs was detected, namely in central nervous system (CNS) and gut. (**E**) Percentage of phenotypes: normal thymus, thymus hyperplasia, and T-ALL at 11 dpf. For each construct injection, we did at least three technical replicates. For detailed statistics, see *Figure 5—source data 1*. Statistical

*Figure 5 continued on next page*

*Figure 5 continued*

data rounded to 4 decimal places. (**F**) Number of pH3 positive cells per thymus counted during confocal imaging. (**G**) Representative confocal images of immunostaining against pH3 (red) and GFP (green). Scale bar: 20 μm. *N* represents the number of biological samples.

The online version of this article includes the following source data and figure supplement(s) for figure 5:

**Source data 1.** Detailed statistics for phenotype comparison using Fisher's exact test and pairwise comparisons; adjusted p-values were calculated with the Benjamini and Hochberg method.

**Figure supplement 1.** Increased *il7* expression upon NICD$^{\Delta PEST}$-mycn overexpression.

or IL7R$^{HI}$ lesioned clones did not show any effect on clonal expansion, but combining delayed differentiation with IL7R$^{DA}$ or IL7R$^{HI}$ doubled proliferation. This result indicates that modulating the effect of NOTCH1 globally does not affect proliferation in our model, which is likely due to feedback inhibition, but specifically targeting the differentiation process leads to clonal expansion.

To test the outcome of the virtual model, we performed a series of in vivo experiments to induce NOTCH1 and IL7R signaling in thymocytes. To constitutively activate NOTCH1 signaling in thymocytes, we took advantage of the previously cloned medaka *notch1b* intracellular domain (NICD) construct (*Aghaallaei et al., 2021*; *Aghaallaei et al., 2022*), which was driven by a thymocyte-specific promoter (*Bajoghli et al., 2015*). Furthermore, the destabilizing PEST domain from the NICD was removed in this construct (hereafter referred to as NICD$^{\Delta PEST}$; *Figure 5C*). This genetic modification was made due to the known impact of the PEST domain on protein stability, as nonsense mutations lacking the PEST domain of the human *NOTCH1* gene have been frequently observed in T-ALL patients (*Breit et al., 2006*; *Ferrando, 2009*; *Weng et al., 2004*; *Figure 5—figure supplement 1A*). To mimic the in silico IL7R$^{HI}$ clones, we employed a previously developed construct wherein a thymocyte-specific promoter drives the medaka full-length *il7r* cDNA (*Aghaallaei et al., 2021*). We then performed a mutation in the extracellular juxtamembrane-transmembrane region of the medaka *il7r* cDNA (*Figure 5—figure supplement 1D*) to develop a dominant active IL7R form, akin to the NPC mutation found in the *IL7R* gene in some T-ALL patients (*Zenatti et al., 2011*; *Oliveira et al., 2022*), hereafter called *il7r*$^{NPC}$. In our experimental setup, the promoter also co-expressed GFP, which allowed us to (i) identify thymocytes expressing the oncogenes, (2) determine the clonal expansion of cells expressing the oncogene, and (3) assess thymus hyperplasia and infiltration into other organs, a characteristic feature for T-ALL. DNA constructs were then injected into blastomeres of embryos at one-cell stage, and they were observed during their development using live imaging, with a focus on two specific phenotypes: First, we assessed whether the thymus exhibited enlargement beyond the normal size for its developmental stage. In particular, thymus lymphoma was defined as hyperplasia with an increase of more than twice the organ's typical size. Second, we identified T-ALL by observing infiltration of GFP-co-expressing malignant cells in other organs, including the brain, intestine, or heart (*Figure 5D*). To align our in vivo experiments with the in silico conditions, we limited our observations to 11 dpf, concluding the experiments at the freshly hatched yolk-sac larval stage. Since thymopoiesis begins at 3 dpf in medaka embryos (*Bajoghli et al., 2015*; *Bajoghli et al., 2009*), this means we monitored thymus growth for a period of 8 days.

None of the embryos injected with the NICD$^{\Delta PEST}$ construct (*N*=32) displayed thymus hyperplasia at 11 dpf (*Figure 5E*), supporting our simulation outcome showing that global activation of NOTCH1 signaling in thymocytes alone is insufficient to result in clonal expansion within a short time frame of 8 days. The MYC oncogene is an endogenous downstream target of NOTCH1 and plays a major role in NOTCH1-induced transformation (*Sanchez-Martin and Ferrando, 2017*). Therefore, to attempt to shift the balance of NOTCH1 action toward proliferation, we decided to introduce the medaka *mycn* cDNA into our construct. Consequently, we observed that 25% of injected embryos with NICD$^{\Delta PEST}$ and *mycn* (*N*=36) displayed thymus hyperplasia at 11 dpf (*Figure 5E*). Among them, 8% also exhibited a massive infiltration of GFP-expressing cells in other organs such as the brain and gut. Further analysis of sorted GFP-expressing thymocytes revealed ectopic *il7* expression, while the expression level of endogenous *il7r* remained unchanged (*Figure 5—figure supplement 1B*). Whole-mount in situ hybridization (WISH) analysis further confirmed a robust upregulation of *il7* expression in the thymus of these embryos (*Figure 5—figure supplement 1C*). MYCN is a basic helix-loop-helix transcription factor that is downstream of several pro-proliferative signaling pathways (*Ruiz-Pérez et al., 2017*), including NOTCH1 in the context of thymocytes (*Sanchez-Martin and Ferrando, 2017*). Together

with our experimental results, it is therefore likely that MYCN could enact the proliferative effect of NOTCH1, shifting the balance in the incoherent feed-forward loop toward proliferation. Interestingly, our results also indicate that transformed thymocytes acquire the ability to release IL-7 (*Figure 5— figure supplement 1B and C*), which may further promote proliferation in an autocrine fashion.

To test whether constitutive IL7R activation could enhance the development of thymus hyperplasia and T-ALL, we injected various constructs to overexpress either *il7r* or the dominant active *il7r*$^{NPC}$ alone, or in combination with NICD$^{\Delta PEST}$ and *mycn*. Thymus hyperplasia was found in a slightly higher frequency of 26% (*N*=49) or 59% (*N*=29) of embryos, respectively, when only *il7r* or *il7r*$^{NPC}$ was overexpressed in thymocytes. Additionally, we FACS-sorted thymocytes and performed a qPCR for lck, since this gene was shown to be upregulated in zebrafish after IL7R$^{DA}$ T-ALL development using RNASeq data (*Oliveira et al., 2022*). Indeed, we detected a significant upregulation of *lck* upon *il7r*$^{NPC}$ overexpression when compared to the WT counterparts. However, upregulation was not significantly different when the WT was compared to the *il7r*$^{HI}$ condition (*Figure 5—figure supplement 1D*). Combining *il7r*$^{NPC}$ and NICD$^{\Delta PEST}$ and *mycn* yielded a notably elevated frequency of ~73% (*N*=33) for thymus hyperplasia (*Figure 5E*). Of this group, 33% also exhibited a T-ALL phenotype, suggesting an additive effect of these factors in the T-ALL development (*Figure 5—source data 1*). GFP and pH3 double staining of embryos exhibiting the T-ALL phenotype further confirmed that the combination of NICD$^{\Delta PEST}$, *mycn*, and *il7r* overexpression in thymocytes resulted in a higher number of mitotically active cells within the thymus (*Figure 5F*). Notably, we observed many pH3-stained cells in the inner zone of the medaka thymus (*Figure 5G*, arrows), an area where WT thymocytes are mitotically quiescent (*Bajoghli et al., 2015*; *Aghaallaei et al., 2021*).

The prediction that higher NOTCH1 signaling does not lead to increased proliferation is likely due to the incoherent feed-forward loop downstream of NOTCH1, which shuts down excessive proliferation by triggering differentiation to a nondividing cell fate (*Figure 5B*). We observed a similar effect in our in vivo experiments, where activation of NOTCH1 alone did not lead to thymus hyperplasia within the observed time window of 8 days (*Figure 5E*). In contrast, combining NOTCH1 with its downstream pro-proliferative effector *mycn* led to thymus hyperplasia. Intriguingly, in other in vivo systems, long-term activation of NOTCH1 signaling in thymocytes is one of the main drivers of T-cell leukemogenesis and T-ALL development (*Weng et al., 2004*; *Lin et al., 2006*; *Chen et al., 2007*; *Liu et al., 2017*; *Neumann et al., 2015*). Several in vivo studies have demonstrated that constitutive activation of the NOTCH1 receptor leads to leukemia development after several weeks in zebrafish (*Chen et al., 2007*; *Blackburn et al., 2012*) or months in mice (*Chiang et al., 2008*; *Hu et al., 2009*; *Sharma et al., 2006*; *Wendorff and Ferrando, 2020*). We did not observe thymus hyperplasia when upregulating NOTCH1 signaling alone, which may stem from the shorter time window of our in vivo experiments (i.e. 8 days). However, in the virtual thymus model, a sole upregulation of NOTCH1 cannot produce thymus hyperplasia regardless of the duration of pathway activation. Different possibilities could reconcile these discrepancies. First, the balance between pro-proliferative and pro-differentiation effects of NOTCH1 in vivo could be different from what we modeled. Second, there could be additional negative feedbacks between proliferation and differentiation, which our model does not consider. Third, proliferation in vivo could continue even past differentiation if NOTCH1 signaling is sufficiently stimulated. Finally, long-term pathway misregulation could act as a stressor that promotes additional compounding lesions.

Overall, our in vivo results support the outcomes of our simulations, showing that overexpression of IL7R alone, but not NOTCH1 alone, is sufficient to induce thymus hyperplasia in a very short time period. However, dual activation of IL7R and the NOTCH1-MYCN axis can promote clonal expansion in vivo, leading to the rapid development of thymus hyperplasia and T-ALL in the medaka model system within 8 days.

## Oversupply of IL-7 by the thymic niche could accelerate the T-ALL development

One of the most unexpected outcomes of our simulations was that a defective TEC network provides lesioned clones with a substantial advantage in proliferation. Given the crucial role of IL-7 in proliferation of thymocytes, and considering that TECs, and not thymocytes, serve as the primary source of IL-7 cytokine in the normal thymic milieu, we next asked whether elevated *il7* expression in TECs alone is sufficient to induce thymus hyperplasia and to enhance T-ALL development in a thymus where

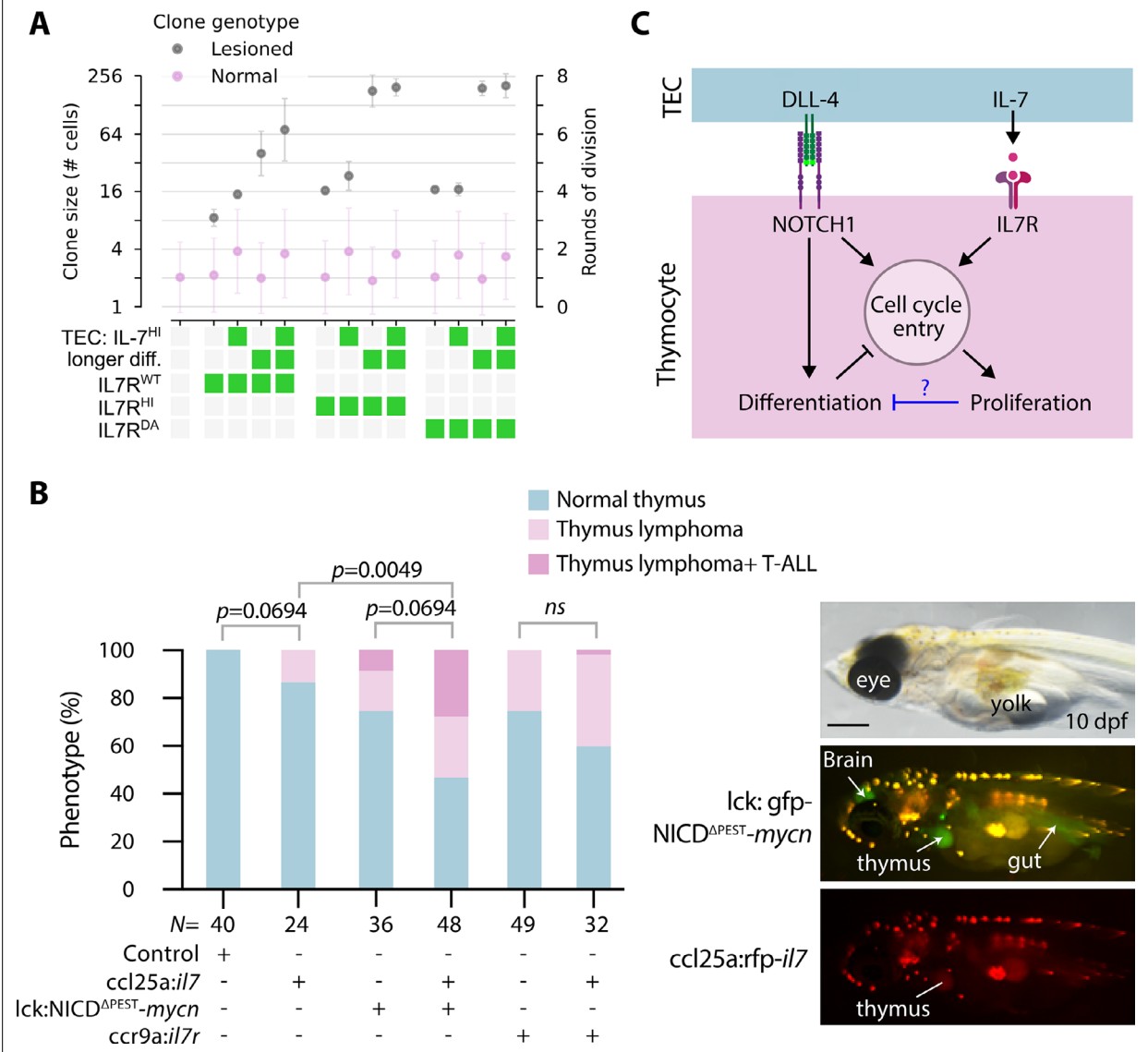

**Figure 6.** Interleukin-7 (IL-7) supplied by the niche can lead to thymus hyperplasia. (**A**) Simulated data; rounds of cell division of normal and lesioned clones for different combinations of lesions and thymic epithelial cells (TECs) expressing IL-7 ubiquitously. Note that some conditions are reproduced from **Figure 5B** for easier comparison. (**B**) Left panel: Frequency of thymus hyperplasia and T-cell acute lymphoblastic leukemia (T-ALL) phenotype observed in the injected embryos at 11 days post-fertilization (dpf). For detailed statistics, see **Figure 5—source data 1**. Statistical data rounded to 4 decimal places. Note that some data from **Figure 5E** are repeated here to facilitate a better comparison. *N* represents the number of biological samples. Right panel: Representative images at 10 dpf after co-injection of lck:gfp-NICD^ΔPEST-*mycn* and ccl25a:tagRFP-il7 constructs. Arrows indicate green fluorescent protein (GFP)-expressing malignant thymocytes in the thymus, brain, and gut. Scale bar: 400 μm. (**C**) We propose a hypothetical negative feedback loop from proliferation to differentiation, which could explain the discrepancy between in silico predictions and in vivo observations.

thymocytes express high levels of *il7r*. This question was also driven by the results of scenario 27, which showed a twofold increase in thymocyte population size when all TECs express IL-7 (**Figure 2E–G**). In this thymic niche enriched with IL-7, lesioned clones either with IL-7^HI or extended differentiation times showed a significant advantage in clonal expansion compared to their WT counterparts (**Figure 6A**). Further, the impact of alterations in the thymic niche on T-ALL development has not been previously addressed. Therefore, we monitored the thymus development of the TEC:IL-7^HI embryos until 11 dpf to mimic scenario 27 in vivo. We found that 12% of these embryos (*N*=24) displayed signs of thymus hyperplasia (**Figure 6B**); however, we did not observe massive infiltration of GFP-expressing cells into other organs. The frequency of thymus hyperplasia was increased to 41% (*N*=32) when DNA constructs designed to overexpress *il7* in TECs were co-injected with the construct overexpressing

*il7r* in thymocytes (*Figure 6B*). Comparably, thymus hyperplasia was only observed in 26% of embryos injected with the IL7R overexpression construct in thymocytes alone (*N*=49). A synergistic effect was observed when *il7* was overexpressed in TEC along with NICD and *mycn* in thymocytes (*Figure 6B*). Interestingly, the frequency of embryos with T-ALL development was increased to 25% (*N*=48) in this group. Taken together, our in vivo results reveal that an excess of IL-7 in the thymic environment, combined with alterations that affect the differentiation status of thymocytes, triggers the T-ALL development in a short period of time.

Overall, several of our experimental observations indicate that there might be additional negative feedback from pro-proliferative signaling to differentiation (*Figure 6C*). Most strikingly, manipulations in IL7R or in thymic IL-7 levels are sufficient to induce hyperplasia and T-ALL (*Figures 5E and 6B*) – an outcome that is much more severe than the model predicts (*Figures 5B and 6A*). In simulations, we could only obtain such massive overproliferation by modifying multiple parameters, including delaying differentiation, accelerating cell divisions, and reducing TEC density (*Figure 4— figure supplement 5*), as differentiation in our model is a hard stop to proliferation regardless of pro-proliferative stimuli. Similarly, the experimental observation that NOTCH1-MYCN upregulation leads to thymus hyperplasia and T-ALL development also suggests that proliferation can proceed unbridled by differentiation in vivo when sufficiently stimulated. A negative feedback loop from pathways regulating proliferation (or cell cycle entry) on differentiation pathways could explain these discrepancies between the virtual thymus model and in vivo observations. We therefore propose that continued thymocyte proliferation can downregulate differentiation pathways and maintain cells in a constant proliferative state.

## Conclusion

This study demonstrates the powerful synergy between computational modeling and in vivo experimental validation in dissecting the complex cellular and molecular interactions within an organ that drive disease initiation and progression. Further, we highlight a previously underexplored area – the role of TECs and the thymic niche in the development of T-ALL – thereby bridging a major gap in our understanding of the earliest stages of disease development.

By simulating over 1500 scenarios, we identified potential drivers such as alterations in the shape, density, and spatial organization of the TEC network – factors that are inherently difficult to manipulate in experimental systems. Consistent with these computational predictions, our in vivo results reveal that an enriched thymic IL-7 milieu can accelerate disease progression, particularly when thymocytes display enhanced IL7R levels or express constitutively active NOTCH1 and MYCN overexpression. These results challenge the traditional gene-centric paradigm of T-ALL and emphasize the critical influence of the spatial structure of the thymic microenvironment.

However, it is worth noting several limitations of our current virtual thymus model. The cell-based computational framework was calibrated to mimic the medaka embryonic thymus (*Bajoghli et al., 2009*; *Bajoghli et al., 2015*; *Aghaallaei et al., 2021*), and as such, it may not fully capture the cellular dynamics or thymic architecture of other species, particularly human. On the technical side, the simulation lacks the ability to represent changes in TEC shape over time and TEC proliferation, as all particles are treated as loose objects and hence movement of a TEC would lead to detachment of its protrusion particles. This limitation could be overcome with a more advanced biomechanical framework that includes bonded interactions between particles such as Tissue Forge (*Sego et al., 2023*), which would ensure that particles belonging to the same cell stay together as one part of the cell moves. Due to this limitation, the dynamic remodeling of the thymic niche is not fully accounted for in our current model. Furthermore, the model simplifies certain T-cell developmental stages as it does not consider cross-regulatory feedback loops between IL-7 and Notch pathways and does not include thymic growth over time. These simplifications are due to the lack of species-specific knowledge and the tools to investigate these questions in medaka. Nonetheless, we anticipate that adapting the framework for use in murine or human thymic contexts, where stage-specific markers and niche components are well characterized and where detailed morphometric studies of TEC architecture are emerging (*Lagou et al., 2024*) would enable us to explore these questions.

Despite its current simplifications, our integrative approach offers valuable insights into how microenvironmental factors influence leukemia expansion and provides a framework for exploring their role in chemoresistance and relapse. Ultimately, these findings open new avenues for developing

therapeutic strategies aimed at disrupting the leukemic-supportive niche, with the potential to improve treatment outcomes in T-ALL.

## Materials and methods
### In silico model

To develop, implement, and simulate the virtual thymus model, we used the modeling and simulation software EPISIM (*Sütterlin et al., 2013*; *Sütterlin et al., 2017*). This multiscale simulation software uses an agent-based paradigm to represent cells as spheres or ellipsoids in three-dimensional space, enabling each of the cells to individually perform internal processes based on flow diagrams and logic rules (e.g. if/else statements) or differential equations, and also implements a partial differential equation solver to simulate diffusion of chemicals in the extracellular space. The virtual thymus model has been comprehensively described in the supplementary material to our previous study (*Aghaallaei et al., 2021*). In this work, we used this model as a baseline to introduce a small number of additions, as explained in the following. A summary of the full model implementation and parameter values is provided in Appendix 1.

### Rate of IL-7 pathway signal transduction

The equation describing the IL-7 signaling rate in each cell was rescaled to introduce the parameter $a_{\text{IL-7}}$ for the IL-7 signal transduction activation rate. The default value of $a_{\text{IL-7}}$ was chosen such that the model behavior did not change. The rescaled equation for IL-7 signal transduction reads:

$$\frac{\mathrm{d}}{\mathrm{d}t}\sigma_{\text{IL-7}} = \left[\text{IL-7R}\right]\left\langle\left[\text{IL-7}_{\text{ex}}\right]\right\rangle a_{\text{IL-7}} - d_{\text{IL-7}}\sigma_{\text{IL-7}} \tag{1}$$

where $\sigma_{\text{IL-7}}$ is the IL-7 pathway signal transduction activity, [IL7R] is the IL7R concentration in the given cell, $\left\langle\left[\text{IL-7}_{\text{ex}}\right]\right\rangle$ is the mean extracellular IL-7 concentration in the cell's microenvironment, and $d_{\text{IL-7}}$ is the signal transduction deactivation rate.

In the scenario where we introduced a lesioned clone with dominant-active IL7R, IL-7 signal transduction activity was calculated as

$$\frac{\mathrm{d}}{\mathrm{d}t}\sigma_{\text{IL-7}} = \left[\text{IL-7R}\right] - d_{\text{IL-7}}\sigma_{\text{IL-7}} \tag{2}$$

Thus, in IL7R clones, neither extracellular IL-7 nor the signal transduction activation rate $a_{\text{IL-7}}$ had an impact on cells' IL-7 signaling pathway activity.

### Depletion of extracellular IL-7

In our original model (*Aghaallaei et al., 2021*), IL-7 ligand is released back to the extracellular space after binding to IL7R. Thus, thymocytes do not affect the extracellular IL-7 gradient, which is given by the partial differential equation

$$\frac{\partial}{\partial t}[\text{IL-7}_{\text{ex}}] = D_{\text{IL-7}}\nabla^2[\text{IL-7}_{\text{ex}}] - k_{\text{IL-7}}[\text{IL-7}_{\text{ex}}] + \sum_i s_i \tag{3}$$

where $D_{\text{IL-7}}$ is the IL-7 diffusion constant, $k_{\text{IL-7}}$ is a baseline level of extracellular IL-7 degradation, and $s_i$ is a source term accounting for IL-7 secretion, e.g., by a subset of TECs. In the modeling platform we use, secretion occurs throughout a cell's volume; more specifically, for solving the partial differential equation, space is discretized into voxels, and all voxels $i$ that intersect with the ellipsoids that are used to represent a cell contribute to the source term.

In this work, we introduce the option, controlled via a Boolean flag *DEPL* that is set by the user before the simulation starts, to simulate a scenario where thymocytes internalize IL-7 ligand after it is bound to IL7R. In this scenario, the extracellular IL-7 concentration is given by:

$$\frac{\partial}{\partial t}[\text{IL-7}_{\text{ex}}] = D_{\text{IL-7}}\nabla^2[\text{IL-7}_{\text{ex}}] - k_{\text{IL-7}}[\text{IL-7}_{\text{ex}}] + \sum_i s_i - \sum_j [\text{IL-7R}]_j \langle[\text{IL-7}_{\text{ex}}]\rangle_j a_{\text{IL-7}} \tag{4}$$

Note that the additional sink term in *Equation 4* represents extracellular IL-7 depletion and is simply the IL-7 signal transduction activity from *Equation 1* summed over all voxels *j* that intersect with thymocytes. We assume that all internalized IL-7 is permanently removed from the extracellular pool.

For lesioned IL7R$^{\text{DA}}$ clones, we used the alternative sink term

$$-\sum_k [\text{IL-7R}]_k \tag{5}$$

### Determining clonal lineages

The EPISIM software we used assigns to each cell a unique cell identifier. We made use of this unique cell identifier to generate unique clonal identifiers as follows. For each thymocyte that entered the simulation via migration from outside the thymus, we used the value of their unique cell identifier to assign the value of a new clonal identifier variable. This cell-specific value of the clonal identifier variable was passed on to any daughter cells produced via cell division. Thus, using the clonal identifier, we essentially generated an in silico clonal label which we used to determine clonally related cells.

### Lesioned clone

To generate a lesioned clone in a given simulation, we created a computational label to mark a single cell selected randomly among new thymic immigrants once the thymic cell population reached its homeostatic size ($t \geq 60$ hr). This label marked this cell and its progeny as a 'lesioned clone', such that we could specifically modulate parameters and introduce new rules only for this single clonal lineage. We tested the effect of several alterations specific to the lesioned clone (*Table 1*).

### Parameter variations

Please refer to Appendix 1 and supplementary material of *Aghaallaei et al., 2021*, for a comprehensive explanation of all parameters and the choice of the reference values. Altered parameters used in this work are listed in *Table 2*.

---

**Table 2.** Parameters varied in this work.
For a full list of parameters and their values, please refer to Appendix 1 and to *Aghaallaei et al., 2021*.

| Symbol | Values tested (reference in bold) | Description |
|---|---|---|
| $N$ | 2; **3**; 4 | Number of concentric subdivisions in algorithm to generate TEC positions. The higher the value, the larger the number of TECs, thus the larger the TEC density. |
| $p_{\text{TEC}}$ | 3; **4**; 5 | Number of protrusions initialized for each TEC. |
| $r_{\text{TEC}}$ | 2.0 μm; **2.5 μm**; 3.0 μm | Radius of the main TEC body. Also scales the radius of the spheres in the TEC protrusions. |
| $DEPL$ | True; **False** | Boolean flag that sets if the simulation has thymocytes that internalize IL-7 according to *Equation 4* in Materials and methods (True) or if it defaults to *Equation 3* in Materials and methods (False). |
| $a_{\text{IL-7}}$ | 120 hr$^{-1}$; **240 hr$^{-1}$**; 480 hr$^{-1}$ | IL7R signaling activation rate |
| $d_{\text{IL-7}}$ | 25 hr$^{-1}$; **50 hr$^{-1}$**; 100 hr$^{-1}$ | IL7R signaling deactivation rate |
| $\theta_{\text{prol}}$ | 1.3; **1.4**; 1.5 | Threshold level of IL-7 and Notch signaling activity required to progress through G1 phase and to commit to the cell cycle. |
| $\mu$ | 5 hr; **7 hr**; 9 hr | Mean cell cycle duration. |
| $T_{\text{diff}}$ | 15 hr; **24 hr**; 33 hr | Minimum duration of the proliferative phase before terminal differentiation. |

---

## Calculation of simulated clone size

Clone size was defined as the number of terminal leaves in a cell lineage tree (see scheme in *Figure 4C*); terminal leaves were counted regardless of the ultimate fate of those cells (i.e. positive or negative selection). All clones of the same genotype (WT or lesioned) belonging to the same simulated scenario were averaged across all replicate simulations of that scenario. This calculation gives the average number of cells per clone in a given scenario. The number of cells was log2-transformed to represent rounds of cell division.

## In vivo model

Medaka (*O. latipes*) husbandry was performed in accordance with the German animal welfare standards (Tierschutzgesetz §11, Abs. 1, Nr. 1, husbandry permit no. 35/9185.46/Uni TÜ). The transgenic line (tg) lck:gfp was described previously (*Bajoghli et al., 2015*). All experiments conducted in medaka embryos were performed prior to the legal onset of animal life stages under protection, utilizing both males and females of the WT line Cab (*Loosli et al., 2000*).

## Cloning of DNA constructs

To overexpress *il7* in TECs, we used the construct ccl25a:*il7*, which was described previously (*Aghaallaei et al., 2021*). To overexpress *il7r* in thymocytes, the full-length medaka *il7r* cDNA (*Aghaallaei et al., 2021*) was cloned into vectors containing a medaka thymocyte-specific promoter (*Bajoghli et al., 2015*) that drives a fluorescent protein (sfGFP or TagRFP). To generate a dominant active form of *il7r*, we introduced three amino acids, asparagine (N), proline (P), and cysteine (C) (known as NPC mutation, as shown in *Zenatti et al., 2011*, and *Oliveira et al., 2022*), in the medaka *il7r* extracellular juxtamembrane-transmembrane interface region after position 266 using site-directed mutagenesis (*Figure 5—figure supplement 1D*), this corresponds to the human NPC mutation in the position 242 of *IL7R* gene. To activate Notch signaling in thymocytes, we utilized the medaka NICD, as described previously (*Aghaallaei et al., 2021*), and removed its PEST domain to enhance protein stability, mimicking the situation reported in most T-ALL patients harboring gain-of-function mutations in the NOTCH1 gene (*Weng et al., 2004*). For overexpressing the *mycn* oncogene, we first isolated the full-length medaka *mycn* cDNA (accession number: ENSORLG00000022362) and introduced a mutation at position 44 from proline to leucine (P44L) using site-directed mutagenesis, as identified in T-ALL patients (*Liu et al., 2017*). The DNA fragments were then cloned into vectors containing a medaka thymocyte-specific promoter, either alone or in combination (*Bajoghli et al., 2015*).

## DNA microinjection

Plasmids at concentration 10–25 ng/µl together with I-*Sce*I meganuclease and NEB buffer (New England BioLabs) were co-injected into the blastomere at one-cell stage embryos. Fluorescent signals based on the constructs sfGFP, tagRFP, or mTurquoise were used to select positive embryos.

## Generation and genotyping of medaka *il7* mutant

The CRISPR-Cas9 approach was employed to generate medaka *il7* crispant. CRISPR RNA (crRNA AGTAGACTGATGCAAAGAAG) was designed using the CCTop website (*Stemmer et al., 2015*; *Stemmer et al., 2017*) and ordered from IDT. RNP complexes were prepared as described before (*Hoshijima et al., 2019*) using Alt-R tracrRNA (IDT) and Alt-R S.p. Cas9 nuclease (IDT). The injection mixture contained a total of 25 µM crRNA:tracrRNA duplex and 25 µM Cas9. Injection was performed into the blastomere at the one-cell stage transgenic embryos carrying the lck:gfp reporter. Injected embryos were raised until 8 dpf and then fixed in 4% PFA/2x PBS+0.1% Tween20 for whole-mount immunostaining. To correlate the phenotype with the genotype, each embryo was genotyped using PCR and Sanger sequencing.

## Phenotype assessment

The development of injected embryos was monitored using a NIKON SMZ18 stereo-fluorescent microscope. Only embryos with fluorescent signals in the thymus region were selected for further analysis. Thymuses of embryos before hatching or freshly hatched yolk sac larvae were examined. Thymuses with at least a twofold increase in size compared to transgenic embryos carrying the lck:gfp reporter construct (*Bajoghli et al., 2015*) were considered as having thymic hyperplasia. The identification of

cells expressing fluorescent proteins and oncogenes outside the thymus, such as in the brain, gut, and heart, was considered indicative of the T-ALL phenotype.

## Whole-mount immunostaining

Immunostaining procedures followed established protocols (*Inoue and Wittbrodt, 2011*; *Aghaallaei et al., 2021*). Briefly, mitotically active cells were identified using a rabbit anti-phosphohistone-3 antibody (Ser10, Millipore 06-570, 1:500 dilution), with a Cy3-donkey anti-rabbit immunoglobulin G secondary antibody (the Jackson Laboratory, 711-165-152; 1:500 dilution). GFP expression was detected using a goat Anti-GFP antibody (Abcam, ab5450, 1:500 dilution), with Alexa 488-donkey anti-goat IgG (Abcam, ab15129, 1:600 dilution) as the secondary antibody. To quantify the number of pH3-expressing cells, the entire thymus region was imaged using an LSM710 (Zeiss) confocal microscope with z-stacks (*z*=1 µm).

## Whole-mount in situ hybridization

WISH in medaka embryos was carried out using digoxigenin-labeled RNA probes, as previously described (*Aghaallaei et al., 2005*). Probes used in this study for *il7r* and *il7* were previously described (*Aghaallaei et al., 2021*).

## Cell sorting and quantitative RT-PCR

Injected embryos were first smashed through a 40 µM strainer (Greiner) and cells were collected in 0.9x PBS with 500 UI Heparinum Natricum (Liquemin). Sorting was then performed using the Cell Sorter MA900 (Sony Biotechnology). WT embryos were used as negative sorting control. Only GFP positive cells were collected and used for RNA isolation. RNA of sorted cells was isolated by using NucleoSpin RNA XS (Macherey-Nagel) and 20 ng carrier RNA, following the manufacturer's protocol. RNA was treated with rDNase and eluted in 14 µl dAqua. The first-strand cDNA synthesis was carried out with random hexamer primers and SuperScript III Reverse Transcriptase (Thermo Fisher Scientific) by following the manufacturer's protocol. SYBR Green Kit (Applied Biosystems) was used for Quantitative PCR on the LightCycler 480 (Roche). The data was evaluated in Microsoft Excel using the ΔCt method and normalized to the housekeeping gene ef1a. The primers used for RT-PCR have been described previously (*Aghaallaei et al., 2021*).

## Statistical analysis

Statistical analysis was performed using RStudio version 2024.4.2.764 (*Posit team, 2024*) using R version 4.4.0 (*R Development Core Team, 2024*), the R library jtools (*Long, 2022*) and GraphPad Prism version 8.0.2. The two-tailed Fisher's exact t-test was used to compare the thymus hyperplasia phenotype with normal thymus. Welch's t-test was used for thymus volume size comparison in WT and *il7* crispants, and a multiple t-test was used for comparison of normal thymus with thymus hyperplasia and T-ALL phenotype. Mann-Whitney test was used to evaluate the qPCRs. p-Values<0.05 were considered as statistically significant. The number of biological samples for each experiment (*N*) is indicated in the figures, and bar graphs present the absolute numbers with mean ± SD, ± SEM, or the percentage as indicated in the figure legends.

## Acknowledgements

The authors thank Anna-Sophia Hellmuth for assistance with *il7* gRNA injection; the Institute of Medical Virology and Microbiology for continuous support in confocal microscopy; Larissa Doll, Julia Skokowa, and Karl Welte for support and encouragement. Initial computational work presented here was performed using the ALICE computer resources provided by Leiden University; later work was performed using the computer lab in the Theoretical Biology and Bioinformatics Department of Utrecht University. This work was supported by the Madeleine Schickedanz Kinderkrebsstiftung (grant number D.30.28666) and Deutsche Forschungsgemeinschaft (BA 5766/5-1). ET is funded by the Dutch Research Council (NWO) in the NWO Talent Programme with project number VI.Veni.222.323. We thank members of the Theoretical Biology and Bioinformatics division for their constructive feedback on the manuscript. We acknowledge support from the Open Access Publishing Funds of the University of Tübingen and the Faculty of Science of Utrecht University for supporting this research.

## Additional information

### Competing interests

Erika Tsingos: Reviewing editor, eLife. The other authors declare that no competing interests exist.

### Funding

| Funder | Grant reference number | Author |
|---|---|---|
| Madeleine Schickedanz Kinderkrebsstiftung | D.30.28666 | Baubak Bajoghli |
| Deutsche Forschungsgemeinschaft | BA 5766/5-1 | Baubak Bajoghli |
| Nederlandse Organisatie voor Wetenschappelijk Onderzoek | VI.Veni.222.323 | Erika Tsingos |

The funders had no role in study design, data collection and interpretation, or the decision to submit the work for publication.

### Author contributions

Erika Tsingos, Conceptualization, Data curation, Software, Formal analysis, Funding acquisition, Investigation, Visualization, Writing – original draft, Writing – review and editing; Advaita M Dick, Formal analysis, Validation, Investigation, Visualization, Methodology, Writing – review and editing; Baubak Bajoghli, Conceptualization, Supervision, Funding acquisition, Visualization, Methodology, Writing – original draft, Project administration, Writing – review and editing, Resources

### Author ORCIDs

Erika Tsingos  https://orcid.org/0000-0002-7267-160X
Baubak Bajoghli  https://orcid.org/0000-0002-7368-7523

### Ethics

Medaka (Oryzias latipes) husbandry was performed in accordance with the German animal welfare standards (Tierschutzgesetz §11, Abs. 1, Nr. 1, husbandry permit no. 35/9185.46/Uni TÜ.). All experiments conducted on medaka embryos took place before the legal start of the stages of animal life that are protected by law.

Reviewer #1 (Public review): https://doi.org/10.7554/eLife.101137.3.sa1
Reviewer #2 (Public review): https://doi.org/10.7554/eLife.101137.3.sa2
Reviewer #3 (Public review): https://doi.org/10.7554/eLife.101137.3.sa3
Author response https://doi.org/10.7554/eLife.101137.3.sa4

## Additional files

### Supplementary files

MDAR checklist

### Data availability

The source code and instructions to run the computational simulation are available on Zenodo. Source data for Figure 4 and its supplementary figures have been made available with the manuscript. Additional materials related to this paper such as plasmids or DNA constructs may be requested from the authors.

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

## Appendix A1

## The virtual thymus model

The cell-based model of the virtual thymus was implemented in the platform EPISIM, which simulates center-based biomechanical interactions for spherical or ellipsoid cells (*Sütterlin et al., 2013*, *Sütterlin et al., 2017*).

A detailed description of the mathematical underpinning and biological inspiration of the model and its parametrization was published previously (*Aghaallaei et al., 2021*). Parameters were adapted from this previous work; a few additional scaling parameters were added to enable differential behavior in lesioned thymocytes (*Appendix 1—Tables 1 and 2*). The code and model files of the most recent version associated with this publication are freely available online on a Zenodo repository (*Tsingos, 2024*). In the following, we recapitulate the main features of the model implementation for the reader's convenience.

The virtual thymus model represents processes in the thymus at the tissue, cellular, and subcellular scales (*Appendix 1—figure 1*). At the tissue scale, we model a thin three-dimensional (3D) slice of the thymus of medaka hatchlings.

At the cellular scale, we model TECs and thymocytes, and both cell types are represented as one or more spherical particles. Biomechanical interactions modeled at the cellular scale are cell motility and cell-cell contact mechanics. This scale also includes processes that change the number of cells such as cell influx into and efflux out of the tissue volume, cell proliferation, and cell death.

At the subcellular scale, we model biochemical interactions such as extracellular diffusion of cytokines, signaling between cells, intracellular signal transduction, and include phenomenological models of the thymocyte cell differentiation and proliferation programs. In the following sections, we provide an overview of each of these processes and refer to original publications for further details where appropriate.

**Appendix 1—table 1.** Parameters defining initial condition.

All parameters were adapted from *Aghaallaei et al., 2021*, unless noted.

| Symbol | Reference value | Description |
|---|---|---|
| $a$ | 50 µm | Long hemi-axis of thymus cylindrical ellipse. |
| $b$ | 25 µm | Short hemi-axis of thymus cylindrical ellipse. |
| $h_c$ | 5 µm | Height of thymus cylindrical ellipse. |
| $g$ | 2 µm | Spatial discretization for solving IL-7 reaction-diffusion equation. |
| $t$ | 15 s | Duration of one simulation step. |
| $r_{TEC}$ | 2.5 µm | Radius of the main TEC body. Also scales the radius of the spheres in the TEC protrusions. |
| $r_C$ | 2 µm | Radius of thymocytes. |
| $N$ | 3 | Number of concentric subdivisions in algorithm to generate TEC positions. The higher the value, the larger the number of TECs, thus the larger the TEC density. |
| $p_{TEC}$ | 4 | Number of protrusions initialized for each TEC. |
| $DEPL$ | False | New parameter introduced in this work. Boolean flag that sets if the simulation operates with thymocytes as IL-7 sinks according to *Equations 10 and 12* (True) or if it defaults to having no sink terms (False). |

**Appendix 1—table 2.** Simulation parameters.
All parameters were adapted from *Aghaallaei et al., 2021*, unless noted.

| Symbol | Reference value | Description |
|---|---|---|
| $\gamma$ | $4 \cdot 10^{-7}$ N·s·µm$^{-1}$ | Friction of the environment. |
| $k_{spawn}$ | 0.45 hr$^{-1}$ | Rate of thymocyte entry into the simulation from two separate positions in the simulation box, resulting in a net influx of 0.9 thymocytes per hour. |
| $\delta_{adh}$ | 1.3 | Scaling factor to define neighborhood and adhesive interaction distances. |
| $\delta_{ol\_max}$ | 0.5 | Scaling factor to define the inner repulsive interaction zone. |
| $\delta_{ol}$ | 0.85 | Scaling factor to define the outer repulsive interaction zone. |
| $d_{ol\_min}$ | 0.1 µm | Defines a neutral zone of no adhesive or repulsive forces. |
| $\mu_{S, imm}$ | 720 µm·hr$^{-1}$ | Mean speed of cells immigrating into or emigrating out of the thymus. |
| $\mu_{S, res}$ | 150 µm·hr$^{-1}$ | Mean speed of cells resident in the thymus |
| $\sigma_S$ | 0.5 | Scaling factor for cell speed variance. |
| $\tau$ | 1 (wild-type, non-resident) $\tau_{diff}$ (wild-type, resident) 0.5 (speed lesion) | Scales the cell speed. In this work, the possibility of including a speed lesion was introduced by setting the parameter to 0.5 for lesioned clones only. |
| $D_{IL-7}$ | $1 \cdot 10^{-12}$ m$^2$·s$^{-1}$ | Extracellular IL-7 diffusion coefficient. |
| $k_{IL-7}$ | 0.0167 hr$^{-1}$ | Extracellular IL-7 degradation rate. |
| $a_{IL-7}$ | 240 hr$^{-1}$ | New parameter introduced in this work. IL7R signaling activation rate (reference value equivalent to previous work). |
| $d_{IL-7}$ | 50 hr$^{-1}$ | IL7R signaling deactivation rate. |
| $k_{aNotch}$ | 0.029 hr$^{-1}$ | Notch signaling deactivation rate. |
| $\kappa$ | 0.3 | Notch-independent differentiation rate. |
| $T_{diff}$ | 24 hr | Minimum duration of the proliferative phase before terminal differentiation. |
| $d_{diff\_lesion}$ | 1 (wild-type) 0.5 (differentiation lesion) | New parameter introduced in this work. Scales the rate of differentiation. |
| $\theta_{diff}$ | 0.4 | Threshold level of IL-7 signaling activity required to differentiate into γδ$^+$ T-cell subtype. |
| $T_{mat}$ | 24 hr | Duration of the maturation phase before thymic selection |
| $\mu$ | 7 hr | Mean cell cycle duration. |
| $k$ | 50 | Shape parameter scaling the variance of the cell cycle distribution function. |
| $t_M$ | 0.5 hr | Duration of the M phase of the cell cycle. |
| $\kappa_S$ | 0.5 | Fraction of the cell cycle duration allocated to the S phase. |
| $\theta_{prol}$ | 1.4 | Threshold level of IL-7 and Notch signaling activity required to progress in G1 phase and to commit to the cell cycle. |

## I. Tissue scale

We define a simulation box of dimensions 220×100×60 μm³. At the center of this box, we define a subregion that represents a slice of the thymus volume as a cylindrical hemi-ellipse with

$$a = 50$$
$$b = 25 \tag{A1}$$
$$h_C = 5$$

where $a$ and $b$ are the elliptical hemi-axes, and $h_C$ is the height of the cylinder representing a slice of the organ; all values are given in micrometers. Furthermore, we subdivide the thymus into two concentric tissue compartments: the inner medulla and the outer cortex (*Appendix 1—figure 1*). The cortex is defined as the region within the outermost 20% of elliptical hemi-axes $a$ and $b$. Since the thymus is a symmetric ellipsoid in young medaka fish, we chose to represent only a slice of the organ to speed up computation and facilitate analysis. The dimensions of the modeled slice were chosen based on measurements of the medaka thymus at 11 dpf (*Bajoghli et al., 2015*).

## II. Tissue scale: initial condition

We initialize the model by distributing TECs in the 3D thymus volume defined by the cylindrical hemi-ellipse in (*Equation A1*). The algorithm has been described previously (*Aghaallaei et al., 2021*). Briefly, we subdivide the thymus volume into a number of subdomains, and in each subdomain, we select a random point with uniform probability; the selected point is then the position of the central sphere representing a TEC's main body. The protrusions of TECs are initialized in four predefined directions starting from the main body with an added small random variation in angle to create irregularity. The radii of TEC protrusion particles are scaled down to generate a gradually tapered protrusion.

## III. Cellular scale: biomechanical interactions

We phenomenologically model passive forces arising from biomechanical contact mechanics, which include intercellular adhesion due to cell-cell adhesion proteins and pressure due to volume exclusion. In addition, we include active forces resulting from cell motility. As the length scale of cells is on the order of 10 μm, we assume low Reynolds numbers and an overdamped environment (*Purcell, 1977*; *Berg, 1993*). Thus, the force-balance equation for a particle $i$ at position $\mathbf{c}_i$ is given by

$$\gamma \frac{\mathrm{d}}{\mathrm{d}t}\mathbf{c}_i = \mathbf{F}_{\text{contact}} + \mathbf{F}_{\text{act}} \tag{A2}$$

where $\gamma$ is the friction or damping constant of the environment, $\mathbf{F}_{\text{contact}}$ is the sum of passive contact forces acting on particle $i$ due neighboring particles in its proximity, and $\mathbf{F}_{\text{act}}$ is the active force exerted by the particle $i$ due to its intrinsic motility. The force balance equation is integrated with an explicit forward Euler scheme.

To simplify the computational implementation, the particles that make up each TEC are fixed in space. In other words, TEC particles are included in the force balance calculation (*Equation A2*), but their position in 3D space is not updated. This enables TECs to act as obstacles to thymocytes but ignores dynamic shape changes and motility of TECs themselves. This simplification is a limitation of the current model.

### Contact forces

The details of the mathematical implementation of contact forces have been published elsewhere (*Sütterlin et al., 2017*). In essence, contact forces are made up of repulsive pressure forces that push particles apart and attractive adhesion forces that pull neighboring particles toward each other. Two particles $i$ and $j$ are considered neighbors if

$$d_{ij} \leq d_{opt}(i,j) \cdot \delta_{adh} \tag{A3}$$

where $d_{ij}$ is the Euclidean distance of the particle centroids, $d_{\text{opt}}(i,j)$ is an optimal distance obtained from the respective dimensions of particle $i$ and its neighbor particle $j$ (for the spherical particle interaction radii that we use in this work the calculation of $d_{\text{opt}}(i,j)$ simplifies to the sum of radii of $i$ and $j$), and $\delta_{\text{adh}} > 1$ is a parameter that defines the neighborhood interaction distance.

The contact forces for each particle-neighbor pair are given by a piecewise function with multiple distance thresholds based on $d_{ij}$ and $d_{opt}(i,j)$, which results in concentric interaction zones (illustrated in *Appendix 1—figure 2*):

$$\mathbf{F}_{\text{contact}} = \begin{cases} \mathbf{F}_{\text{pr,exp}} & \text{if} & 0 & < d_{ij} < d_{opt}(i,j) \cdot \delta_{\text{ol}} \cdot \delta_{\text{ol\_max}} \\ \mathbf{F}_{\text{pr, lin}} & \text{if} & d_{\text{opt}}(i,j) \cdot \delta_{\text{ol}} \cdot \delta_{\text{ol\_max}} & \leq d_{ij} < d_{opt}(i,j) \cdot \delta_{\text{ol}} - d_{\text{ol\_min}} \\ \mathbf{0} & \text{if} & d_{\text{opt}}(i,j) \cdot \delta_{\text{ol}} - d_{\text{ol\_min}} & \leq d_{ij} < d_{\text{opt}}(i,j) + d_{\text{ol\_min}} \\ \mathbf{F}_{\text{adh}} & \text{if} & d_{\text{opt}}(i,j) + d_{\text{ol\_min}} & \leq d_{ij} < d_{\text{opt}}(i,j) \cdot \delta_{\text{adh}} \\ \mathbf{0} & \text{if} & d_{\text{opt}}(i,j) \cdot \delta_{\text{adh}} & \leq d_{ij} \end{cases} \quad (A4)$$

Here, $\delta_{\text{ol\_max}}$, $\delta_{\text{ol}}$, $d_{\text{ol\_min}}$ are parameters that scale the interaction distances. The parameter $\delta_{\text{ol\_max}}$ $\in (0, 1)$ defines the inner repulsive interaction zone where the repulsion force $\mathbf{F}_{\text{pr,exp}}$ exponentially increases as $d_{ij}$ decreases; $\delta_{\text{ol}} \in (0, 1)$ defines the outer repulsive interaction zone where the repulsion force $\mathbf{F}_{\text{pr,lin}}$ linearly increases as $d_{ij}$ decreases; $d_{\text{ol\_min}} \in (0, 1)$ defines a neutral zone with no net force; $\delta_{\text{adh}}$, which is used to determine neighborhood in *Equation A3* additionally defines the interaction zone where adhesive forces $\mathbf{F}_{\text{adh}}$ dominate (*Appendix 1—figure 2*). For the functional form chosen for the force terms, we refer to *Sütterlin et al., 2017*. Based on previous work, the values $\delta_{\text{ol\_max}}$ = 0.5; $\delta_{\text{ol}}$ = 0.85; $d_{\text{ol\_min}}$ = 0.1 μm; $\delta_{\text{adh}}$ = 1.3 were chosen as they result in appropriate spacing of particles without mechanical instability (*Sütterlin et al., 2017*, *Tsingos et al., 2019*; *Aghaallaei et al., 2021*). Note that our approach is consistent with state-of-the-art models of force-based interactions between particles to simulate cells (*Pathmanathan et al., 2009*; *Osborne et al., 2017*).

## Active forces

During the simulation, each cell determines an active instantaneous displacement $\mathbf{d}_{\text{act}}$ from its instantaneous speed $s$ and a unit vector $\mathbf{v}$ representing cell orientation

$$\mathbf{d}_{\text{act}} = s \cdot \mathbf{v} \cdot \text{d}t \quad (A5)$$

where $t$ is time. Both $s$ and $\mathbf{v}$ are functions of the cell's internal state and its position in the tissue and are dynamically evaluated during the simulation. For an in-depth explanation, see the supplementary material of *Aghaallaei et al., 2021*. In brief, the functional form for the speed $s$ was chosen as

$$s = \mu_S \cdot \left(1 + \sigma_S \text{X}\right) \cdot \tau \quad (A6)$$

where $\mu_S$ and $\sigma_S$ are model parameters, X $\in$ (–1, 1] is a random uniform number, and $\tau$ is either set to $\tau$=1 for thymocytes entering or exiting the thymus, or set to $\tau = \tau_{\text{diff}}$ for thymocytes currently residing in the thymus (for an explanation on $\tau_{\text{diff}}$, see section *IX. Subcellular scale: differentiation model*). Essentially, setting $\tau = \tau_{\text{diff}}$ enables speed to be linearly increasing with developmental stage, which is an assumption we made based on experimental measurements of thymocyte motility in vivo (*Bajoghli et al., 2015*). For thymocytes of lesioned clones with reduced cell speed, we set $\tau$=0.5. Coefficients $\mu_S$ and $\sigma_S$ were fit to the speed measured in confocal imaging data of thymocytes in medaka; different coefficients for $\mu_S$ were used for thymic immigrant/emigrants and thymic resident cells (*Aghaallaei et al., 2021*). Cells from a lesioned clone with slower cell speed had the value of their $\mu_S$ coefficient halved compared to WT cells.

The orientation vector $\mathbf{v}$ is a unit vector obtained by combining a directional vector $\mathbf{u}$ pointing from the current cell position toward a target location and a random vector $\mathbf{w}$ that introduces a random bias. The directional $\mathbf{u}$ components and random $\mathbf{w}$ components are weighted differently depending on cell location, developmental stage, and each cell's intrinsic expression levels of the cytokine receptor *Ccr9b* (schematically summarized in *Appendix 1—figure 3*). For a detailed explanation of the implementation, choice of parameter values, and a sensitivity analysis of tissue homeostasis with respect to parameters, we refer to our previous publication (*Aghaallaei et al., 2021*).

The active displacement $\mathbf{d}_{\text{act}}$ is used to obtain the cell motility force $\mathbf{F}_{\text{act}}$

$$\mathbf{F}_{\text{act}} = \mathbf{d}_{\text{act}} \cdot \gamma \frac{\text{d}}{\text{d}t} \quad (A7)$$

where $\gamma$ is the friction or damping constant of the environment. In the implementation, the active force $\mathbf{F}_{act}$ obtained in the previous simulation step with (*Equation A7*) is used in the equation of motion (*Equation A2*) at the beginning of each new simulation step. Because contact forces $\mathbf{F}_{contact}$ to neighboring cells may hinder cell motility, the effective displacement of cells that results from *Equation A2* may differ from the displacement calculated in *Equation A5*.

## IV. Cellular scale: cell addition and removal

Cell addition can occur via influx or cell proliferation, and cell removal by efflux or cell death. Note that these mechanisms apply exclusively to thymocytes. The number of TECs is immutable after initialization in the simulation. This simplification is a limitation of the current model.

### Cell addition

The model includes two mechanisms for adding new cells to the simulated volume: (1) influx of early undifferentiated thymocytes from areas elsewhere in the body and (2) increase in cell number due to thymocyte proliferation.

In medaka, thymocytes enter the organ from the ventral side at an estimated influx rate of 9 thymocytes per hour (*Bajoghli et al., 2015*). The model implements cell influx by instantiating new cells at two positions at the bottom of the simulation box with rate $k_{spawn}$. Since the model represents about one-tenth of the full organ's volume, we scaled down the influx rate accordingly to $k_{spawn} = 0.45$ thymocytes per hour, which adds up to 0.9 thymocytes per hour.

Cell proliferation follows a model of the cell division cycle, where progression through G1 phase and commitment to cell division depends on intracellular signaling at the subcellular level (see section *X. Subcellular scale: cell proliferation model* for an in-depth explanation). Once the subcellular program of the cell division cycle is complete, the implementation instantiates a new cell adjacent to the cell that completed the cycle at a random orientation; this simulates the close positioning of daughter cells just after cytokinesis with a random spindle orientation.

### Cell removal

Two processes account for cell removal from the simulation: (1) efflux of fully differentiated thymocytes that are positively selected, (2) death of fully differentiated thymocytes that are negatively selected. To our knowledge, there is no evidence of substantial cell death at earlier stages of differentiation in medaka (*Bajoghli et al., 2015*); hence, we do not include other mechanisms of cell removal.

Once fully differentiated, each thymocyte undergoes thymic selection (see subsection *IX. Subcellular scale: differentiation model* for details). Positively selected thymocytes migrate out of the thymus and eventually leave the simulation box, while negatively selected thymocytes stay in position and slowly shrink until their radius reaches 25% of the original value, at which point they are removed from the simulation.

## V. Subcellular scale: extracellular diffusion of IL-7

The extracellular IL-7 concentration $[\text{IL-7}]_{ex}$ follows the reaction-diffusion equation

$$\frac{\partial}{\partial t}\left[\text{IL-7}_{ex}\right] = D_{\text{IL-7}}\nabla^2\left[\text{IL-7}_{ex}\right] - k_{\text{IL-7}}\left[\text{IL-7}_{ex}\right] + \sum_i s_{\text{SOURCE},i} - \sum_i s_{\text{SINK},i} \tag{A8}$$

where $D_{\text{IL-7}}$ is the diffusion coefficient of extracellular IL-7, $k_{\text{IL-7}}$ is the baseline extracellular degradation of IL-7, and $s_{\text{SOURCE}}$ is a source term summed over the volume of cells that secrete IL-7 and $s_{\text{SINK}}$ is a sink term summed over the volume of cells that consume extracellular IL-7. The diffusion coefficient $D_{\text{IL-7}} = 1 \cdot 10^{-12}$ was chosen based on reported diffusion coefficients for cytokines (*Moghe et al., 1995*) and the baseline degradation $k_{\text{IL-7}} = 0.0167$ per hour was chosen by parameter scan to produce a gradient that decays within a few cell diameters as reported for other cytokine gradients (*Thurley et al., 2015*). In the computational implementation, the parameter value for the secretion rate $s_{\text{SOURCE}}$ of IL-7-secreting TECs is set to 1 concentration unit per voxel of the numerical discretization. This choice ensures that the extracellular IL-7 gradient stays within the interval range of [0, 1] in unscaled concentration units. IL-7-secreting TECs include all TECs whose main cell body is located in the cortex tissue subcompartment, which results in an intra-thymic gradient of IL-7. Thymocytes with a lesion inducing autocrine IL-7 also acted as sources. In the baseline model of *Aghaallaei et al., 2021*,

the sink term in *Equation A8* is set to zero, while in the present work, we compared zero to nonzero sink terms for all thymocytes.

The initial condition used to solve the partial differential equation in *Equation A8* is chosen as uniform zero concentration, and the boundary conditions at the edges of the simulation box are set to a constant zero concentration. In test simulations, we verified that the gradient achieves its numerical steady state within as few as 10 simulation steps, which corresponds to 150 s, and that the simulation box is sufficiently large to prevent boundary artifacts (*Aghaallaei et al., 2021*).

## VI. Subcellular scale: intracellular IL-7 signal transduction

The model for intracellular IL-7 signal transduction is summarized in *Appendix 1—figure 4*. For WT thymocytes, the intracellular IL-7 pathway signal transduction level $\sigma_{\text{IL-7}}$ is governed by

$$\frac{\text{d}}{\text{d}t}\sigma_{\text{IL-7}} = [\text{IL-7R}] \left\langle [\text{IL-7}_{\text{ex}}] \right\rangle a_{\text{IL-7}} - d_{\text{IL-7}}\sigma_{\text{IL-7}} \tag{A9}$$

where [IL-7R] is the concentration of IL-7 receptor of the cell, $\left\langle [\text{IL-7ex}] \right\rangle$ is the average extracellular IL-7 concentration in the cell's microenvironment (defined as the voxels of the numerical discretization that overlap with the cell's radius), $a_{\text{IL-7}}$ is the pathway activation rate, and $d_{\text{IL-7}}$ is the pathway deactivation rate. In simulations where we used a sink term in the reaction-diffusion equation for extracellular IL-7 (*Equation A8*), the sink term for WT cells was given by the first term of (*Equation A9*):

$$s_{\text{SINK}} = [\text{IL-7R}] \left\langle [\text{IL-7}_{\text{ex}}] \right\rangle a_{\text{IL-7}} \tag{A10}$$

In WT cells, the concentration of IL-7 receptor [IL-7R] was set to a random value in the range [0, 1) in unscaled concentration units. This amount of IL-7 receptor was set at the moment the cell entered the simulation via cell influx and was inherited clonally to daughter cells. In previous work, we examined the impact of this modeling choice on proliferation and differentiation, both of which depend on IL-7 signaling. In short, clones with higher IL-7 tended to proliferate more and differentiate into γδ⁺ T-cell subtypes, while clones with lower IL-7 tended to proliferate less and differentiate into αβ⁺ T-cell subtypes (*Aghaallaei et al., 2021*). The model does not include any dynamic feedback mechanisms acting on the receptor concentration, which is a limitation that could be addressed in future work.

In this work, we introduced lesioned cells with impaired IL-7 signaling. If the lesion involved expressing a dominant-active IL-7R, the following alternative functional form for the pathway activation $\sigma_{\text{IL-7}}$ was used

$$\frac{\text{d}}{\text{d}t}\sigma_{\text{IL-7}} = [\text{IL-7R}] - d_{\text{IL-7}}\sigma_{\text{IL-7}} \tag{A11}$$

Correspondingly, the sink term for these lesioned cells was formulated as

$$s_{SINK} = [\text{IL-7R}] \tag{A12}$$

The level of IL-7 receptor expression of cells with a dominant-active receptor lesion was set to 1 unscaled concentration unit, which is the maximum attainable in the WT population.

Cells with an overexpression lesion used *Equation A9* and *Equation A10* like WT cells, but their IL-7 receptor level was set to 10 unscaled concentration units, which is 10-fold the maximum attainable in the WT population.

## VII. Subcellular scale: intercellular Notch signaling

Based on expression patterns in medaka fish thymus (*Bajoghli et al., 2009*), all TECs in the model express the Notch ligand Dll-4a, while all thymocytes express the Notch receptor Notch1b. Because concentrations of ligand and receptor on the cell membrane are not known, we set the concentration of Dll-4a on TECs to [DLL4a]=1 unscaled concentration units and the level of Notch1b receptor on thymocytes to [Notch1b]=1 unscaled concentration units. We do not include feedback mechanisms that dynamically regulate the expression levels of the ligand or the receptor; this is a limitation of the model that could be explored in future work.

Since Notch signaling is mediated by direct cell-cell contact, we used the cell-neighbor pair information obtained from the cellular scale biomechanical interaction model (*Equation A3*) to determine if a ligand-expressing TEC was in contact with a receptor-expressing thymocyte.

## VIII. Subcellular scale: intracellular Notch signal transduction

The model for intracellular Notch signal transduction is summarized in *Appendix 1—figure 4*. The activity of the intracellular Notch signaling pathway is modeled as follows:

$$\frac{\mathrm{d}}{\mathrm{d}t}\sigma_{\mathrm{Notch}} = \begin{cases} 1 & \text{if at least one neighbour is a DLL-4-expressing TEC} \\ -\kappa_{\sigma_{\mathrm{Notch}}}\sigma_{\mathrm{Notch}} & \text{otherwise} \end{cases} \tag{A13}$$

where $k_{\sigma\mathrm{Notch}}$ is a parameter for the pathway deactivation rate which we fitted to experimental data of pharmacological inhibition of the pathway in medaka (*Saturnino et al., 2018*). The pathway's maximum activity level is set to 1 unscaled unit. Thymocytes that have a dominant-active lesion of the Notch pathway always have their pathway activity set to 1.

## IX. Subcellular scale: differentiation model

Thymocytes entering the thymus start out as early thymic progenitors in an undifferentiated state; after some time, they can differentiate into either $\alpha\beta^+$ or $\gamma\delta^+$ T-cell subtypes (*Appendix 1—figure 5*). To model this process, each cell has an internal variable $\tau_{\mathrm{diff}}$ that tracks its differentiation progress. A new cell that immigrates into the thymus starts with $\tau_{\mathrm{diff}} = 0$; the value of $\tau_{\mathrm{diff}}$ is then increased at every simulation step, and when $\tau_{\mathrm{diff}} \geq 1$ for an individual cell, then this cell's differentiation is considered complete. When a cell divides, both daughter cells inherit the mother cell's value of $\tau_{\mathrm{diff}}$. Experimental observations showed that Notch signaling stimulates differentiation (*Aghaallaei et al., 2021*). To account for these observations, we model the rate of differentiation (i.e. the rate of increase of $\tau_{\mathrm{diff}}$) with

$$\frac{\mathrm{d}}{\mathrm{d}t}\tau_{\mathrm{diff}} = \frac{\kappa + \sigma_{\mathrm{Notch}}(1-\kappa)}{T_{\mathrm{diff}}} \cdot d_{\mathrm{diff\_lesion}} \tag{A14}$$

where $\sigma_{\mathrm{Notch}}$ is the Notch pathway activity given by *Equation A3*, $\kappa \in [0, 1]$ is a parameter for the rate of Notch-independent differentiation, which we previously estimated at 0.3, and $T_{\mathrm{diff}}$ is a parameter for the minimum duration of differentiation, which we previously estimated at 24 hr (*Aghaallaei et al., 2021*). The parameter $d_{\mathrm{diff\_lesion}}$ was set to 1 for unlesioned WT thymocytes and to 0.5 for thymocytes with a lesion leading to slower differentiation.

The fate selection between $\alpha\beta^+$ or $\gamma\delta^+$ T-cell subtypes depends on a threshold level of IL-7 signaling at the moment of differentiation (i.e. when $\tau_{\mathrm{diff}}$ first exceeds 1):

$$\begin{aligned} &\text{if} & \sigma_{\mathrm{IL\text{-}7}} \geq \theta_{\mathrm{diff}} & \quad \text{differentiate into } \gamma\delta^+ \\ &\text{else} & & \quad \text{differentiate into } \alpha\beta^+ \end{aligned} \tag{A15}$$

In previous work, we studied the effect of altering this threshold level and identified $\theta_{\mathrm{diff}} = 0.4$ as giving a good fit to the WT situation (*Aghaallaei et al., 2021*).

Differentiated $\alpha\beta^+$ and $\gamma\delta^+$ cells linger in the thymus until they mature. Similar to differentiation, we model the maturation process phenomenologically with

$$\frac{\mathrm{d}}{\mathrm{d}t}\tau_{\mathrm{mat}} = \frac{1}{T_{\mathrm{mat}}} \tag{A16}$$

where $\tau_{\mathrm{mat}}$ is initially set to 0 and tracks the maturation progress and $T_{\mathrm{mat}}$ is a parameter for the minimum duration of maturation, which we previously estimated at 24 hr (*Aghaallaei et al., 2021*).

When $\tau_{\mathrm{mat}} \geq 1$, maturation is considered complete. Once mature, cells are either negatively selected and hence undergo apoptosis or positively selected and leave the thymus, exiting the simulation.

## X. Subcellular scale: cell proliferation model

The cell proliferation model is summarized in *Appendix 1—figure 6*. Thymocytes in the model are only competent for proliferation if they are undifferentiated ($\tau_{\mathrm{diff}} < 1$). Each cell draws a random

Erlang-distributed random number $E$ with mean $\mu=7$ hr and shape value $k=50$. This random number is then set as the minimum duration of the next cell cycle – i.e., the fastest possible cell cycle if sufficient pro-proliferative signals are present. We subdivide the randomly chosen cell cycle interval $E$ into four phases using the assumption that a cell spends $\kappa_S = 50\%$ of the cell cycle duration in the S phase and $t_M = 30$ min in the M phase, as follows:

$$
\begin{aligned}
t_{G1} &= \frac{E}{2}\left(1 - \frac{\kappa_S}{2}\right) \\
t_S &= E\kappa_S \\
t_{G2} &= \frac{E}{2}\left(1 - \frac{\kappa_S}{2}\right) - t_M \\
t_M &= t_M
\end{aligned}
\tag{A17}
$$

where $t_{G1}$ is the time spent in G1 phase, $t_S$ the time spent in S phase, $t_{G2}$ the time spent in G2 phase. The model also includes a G0-like quiescence due to the absence of sufficient pro-proliferative signals. At the beginning of a new cycle, each cell sets the value of its internal variable $\tau_{cycle} = 0$, which tracks the progression through the cell cycle. This variable is incremented as

$$
\frac{\mathrm{d}}{\mathrm{d}t}\tau_{cycle} =
\begin{cases}
0 & \text{if } \tau_{cycle} \leq t_{G1} \text{ and } \sigma_{IL\text{-}7} + \sigma_{Notch} > \theta_{prol} \\
\frac{1}{E} & \text{otherwise}
\end{cases}
\tag{A18}
$$

where $\sigma_{IL\text{-}7}$ is the IL-7 pathway signaling given by *Equation A9* (or *Equation A11* for clones with dominant-active receptor lesion), $\sigma_{Notch}$ is the Notch pathway signaling given by *Equation A13*, and $\theta_{prol} = 1.4$ is a threshold parameter.

When $\tau_{cycle} \geq E$, the cell divides into two daughter cells, and for both daughters, the value is reset to $\tau_{cycle} = 0$. Essentially, *Equation A18* models how the IL-7 and Notch pathways act as independent permissive pro-proliferative signals on G1 phase progression; if these signals are absent or too weak, then the cell cycle is delayed as $\tau_{cycle}$ does not increase and the cell remains in a G0-like quiescent state.

During the initial creation of the model, the sensitivity of the model with respect to the parameter values in *Equation A17 and Equation A18* was studied by parameter scan (*Aghaallaei et al., 2021*). To choose appropriate values for the mean cell cycle duration $\mu$ and the threshold proliferation-permissive level $\theta_{prol}$, the cell population size and the percentage of simulated mitotic events at homeostasis were compared to experimental measurements of pH3 staining from confocal slices (*Aghaallaei et al., 2021*).

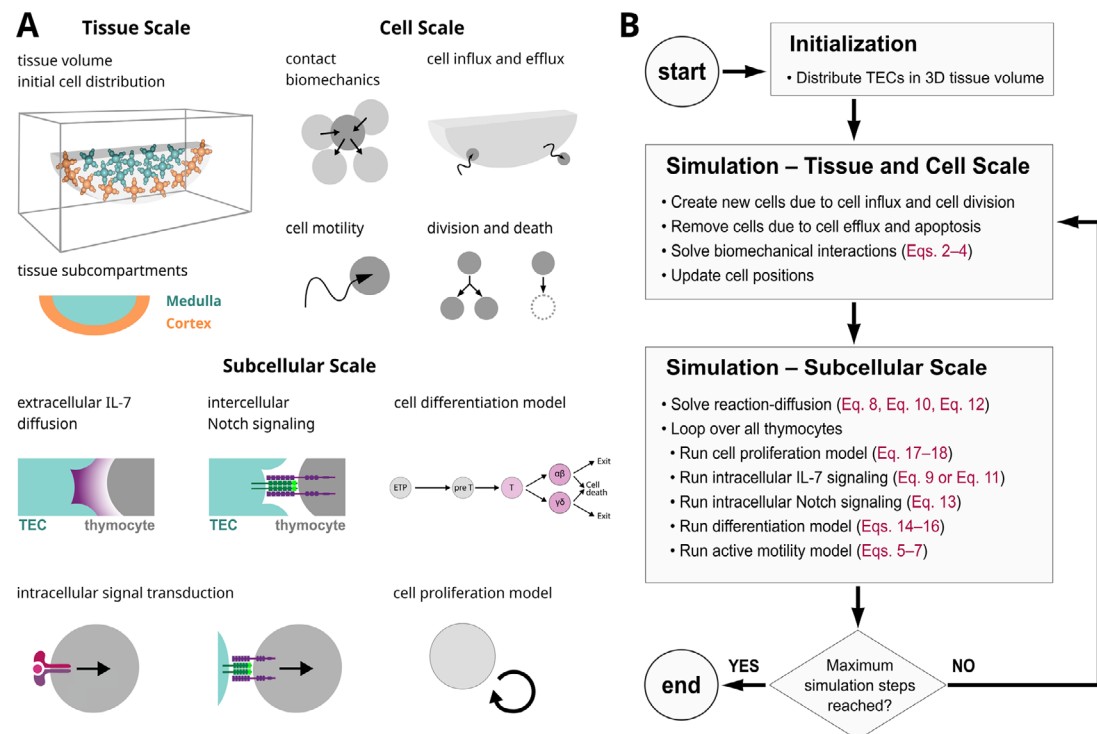

**Appendix 1—figure 1.** Overview of simulation components. (**A**) Schematic representation of model components at the tissue, cell, and subcellular scale. (**B**) Flowchart illustrating the model operation with references to equations shown in the text.

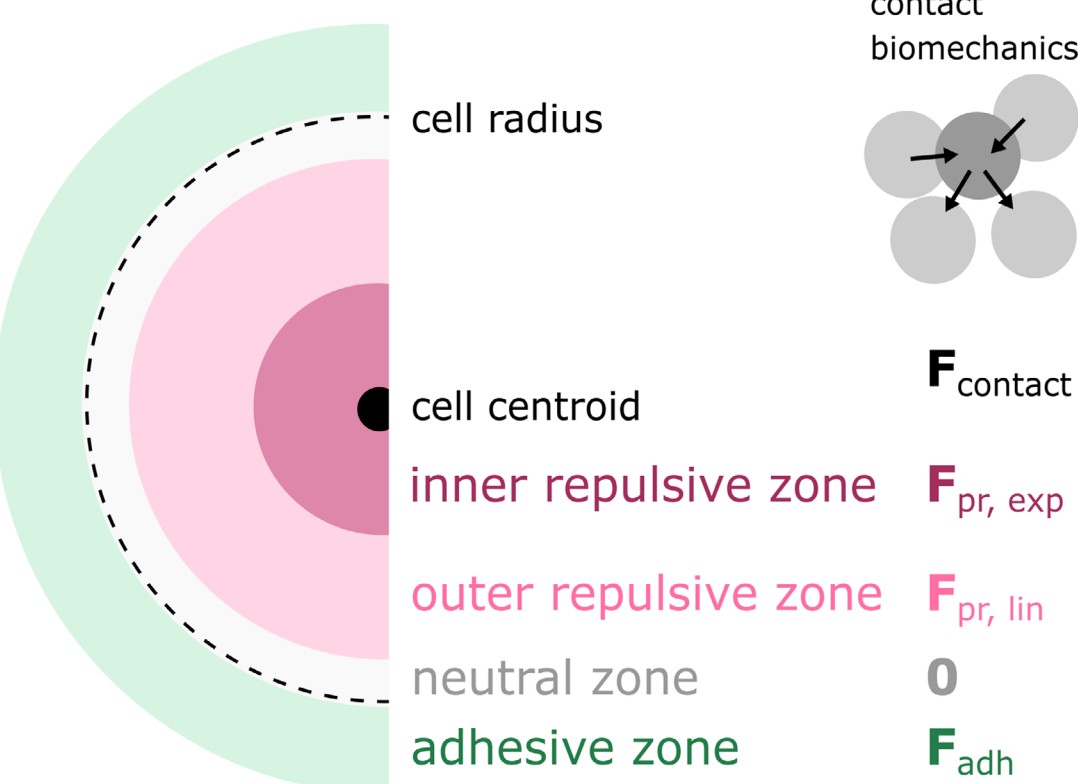

**Appendix 1—figure 2.** Biomechanical interaction zones. Schematic representation of the different interaction zones that are used for calculating biomechanical contact forces. The radii of the interaction zones in the image have been scaled assuming two spherical particles of identical radius.

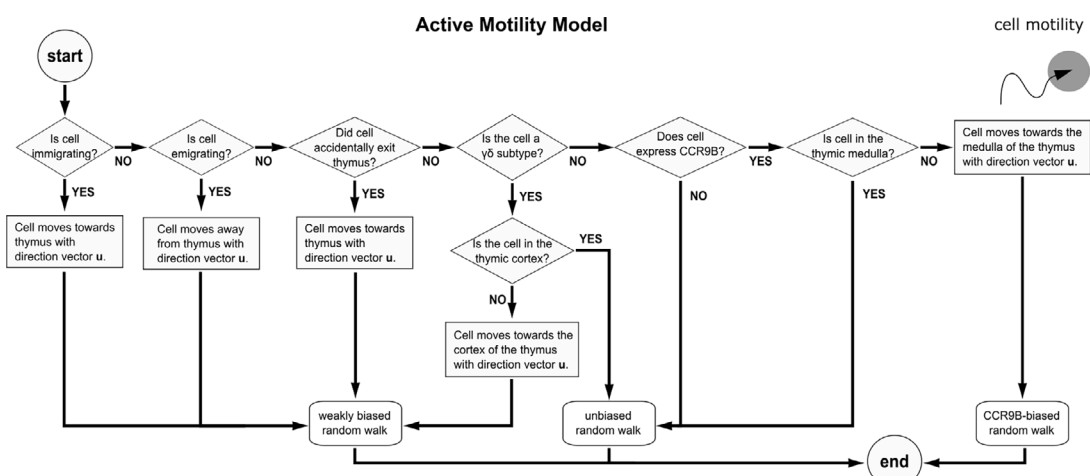

**Appendix 1—figure 3.** Active motility model. Schematic flowchart illustrating how cell location and state affect cell motility. Early thymic progenitors (ETPs) entering the thymus for the first time ('immigrating'), mature thymocytes leaving the thymus after positive selection ('emigrating'), and thymocytes at various stages of development that accidentally exit the thymus volume ('accidentally exit') have a strong directed component and weak random component to their migration. Otherwise, resident thymocytes at all stages of development do a random walk if they are in the correct thymic compartment, or a biased random walk if they are not in the correct thymic compartment.

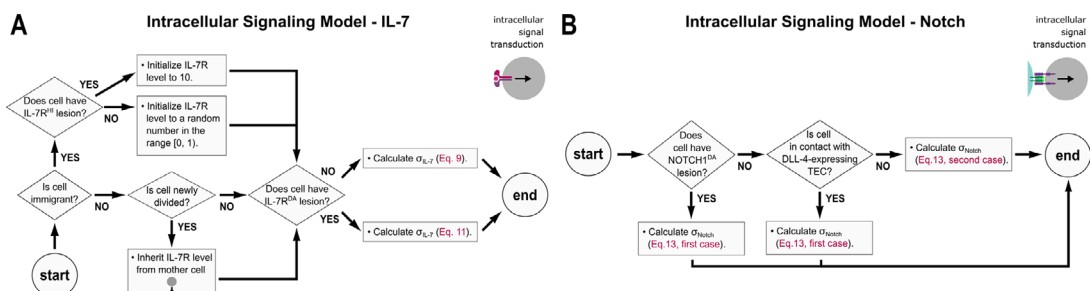

**Appendix 1—figure 4.** Intracellular signal transduction model. (**A**) Schematic flowchart illustrating the intracellular IL-7 signal transduction model. Early thymic progenitors (ETPs) entering the thymus for the first time are labeled as 'immigrants' and undergo an initialization of variables. (**B**) Schematic flowchart illustrating the intracellular Notch signaling model.

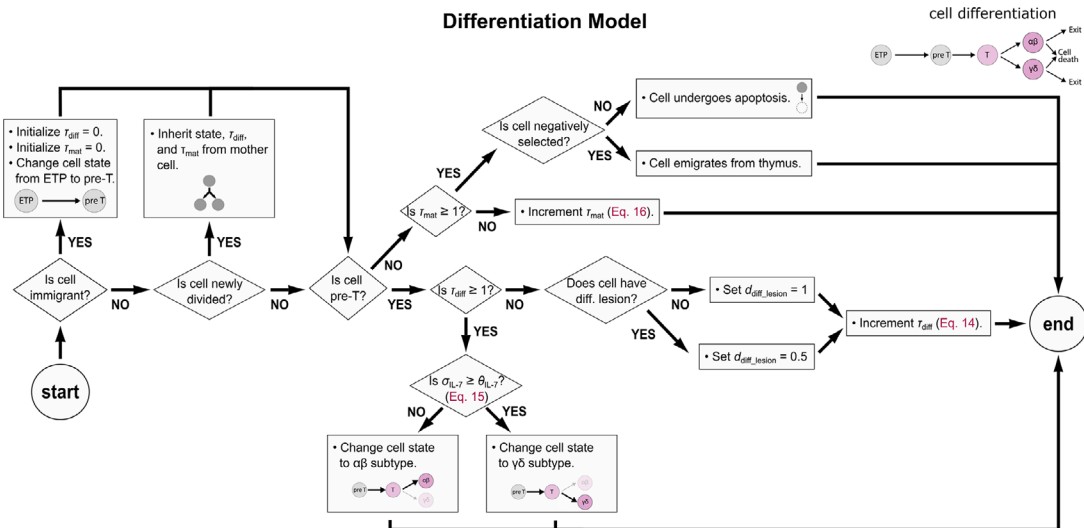

**Appendix 1—figure 5.** Differentiation model. Schematic flowchart illustrating how cell differentiation stage is updated at every simulation step. Early thymic progenitors (ETPs) entering the thymus for the first time are labeled as 'immigrants' and undergo an initialization of variables.

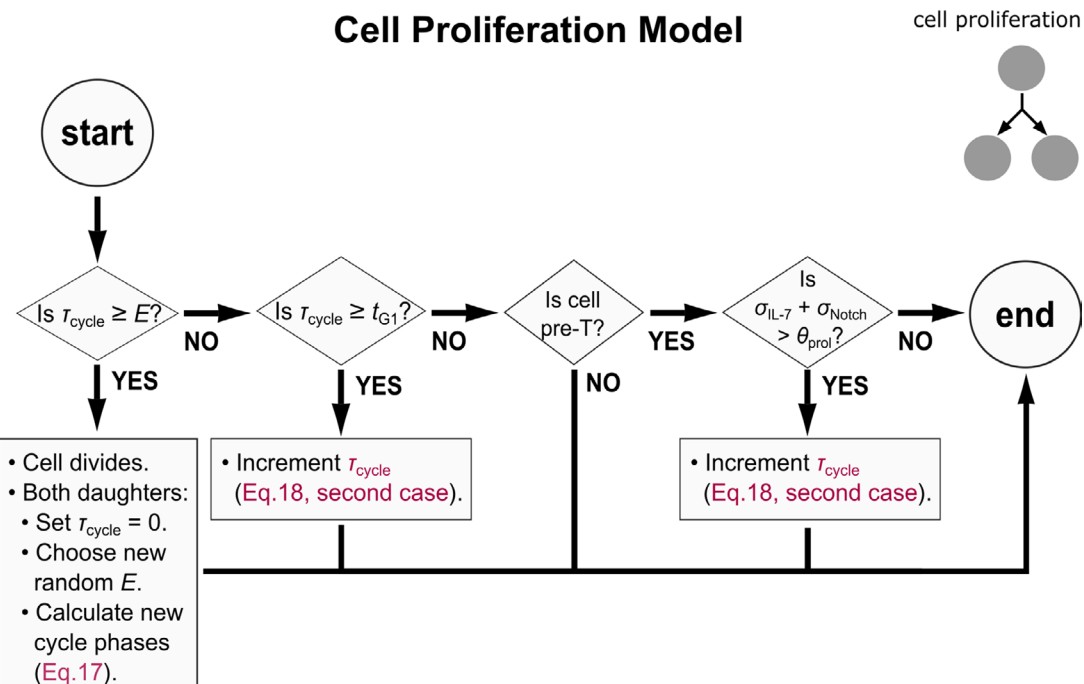

**Appendix 1—figure 6.** Proliferation model. Schematic flowchart illustrating the cell proliferation model. At each simulation step, cells first check if they completed the cell cycle and divide if yes. Next, if a cell is already past the G1 phase, it is committed to finish the cell cycle. Note that this condition is checked before the cell state, hence a differentiated cell could divide once if it differentiated while already committed to the cell cycle. Only undifferentiated cells ('pre-T') can start a new cell cycle, which is conditional on sufficient pro-proliferative IL-7 and Notch signals.

