## [Editor Report · eLife Assessment]

This **important** study combines agent-based modelling and in vivo experiments in medaka embryos to provide new insights into the role of the thymic niche in T cell development. The modelling yields some interesting and **solid** findings regarding the importance of thymic epithelial cells. This study would be of interest to oncologists, immunologists, and mathematical modelers.

---

## [Referee Report · Reviewer #1 (Public review)]

Summary:

This study uses a cell-based computational model to simulate and study T cell development in the thymus. They initially applied this model to assess the effect of the thymic epithelial cells (TECs) network on thymocyte proliferation and demonstrated that increasing TEC size, density, or protrusions increased the number of thymocytes. They postulated and confirmed that this was due to changes in IL7 signalling and then expanded this work to encompass various environmental and cell-based parameters, including Notch signalling, cell cycle duration, and cell motility. Critical outcomes from the computational model were tested in vivo using medaka fish, such as the role of IL-7 signalling and minimal effect of Notch signalling.

Strengths:

The strength of the paper is the use of computational modelling to obtain unique insights into the niche parameters that control T cell development, such as the role of TEC architecture, while anchoring those findings with in vivo experiments. I can't comment on the model itself, as I am not an expert in modelling, however, the conclusions of the paper seem to be well-supported by the model.

---

## [Referee Report · Reviewer #2 (Public review)]

Summary:

The authors have worked up a ``virtual thymus' using EPISIM, which has already been published. Attractive features of the computational model are stochasticity, cell-to-cell variability, and spatial heterogeneiety. They seek to explore the role of TECs, that release IL-7 which is important in the process of thymocyte division.

In the model, ordinary clones have IL7R levels chosen from a distribution, while `lesioned' clones have an IL7R value set to the maximum. The observation is that the lesioned clones are larger families, but the difference is not dramatic. This might be called a cell-intrinsic mechanism. One promising cell-extrinsic mechanism is mentioned: if a lesioned clone happens to be near a source of IL-7 and begins to proliferate, the progeny can crowd out cells of other clones and monopolise the IL-7 source. The effect will be more noticeable if sources are rare, so is seen when the TEC network is sparse.

Strengths:

Thymic disfunctions are of interest, not least because of T-ALL. New cells are added, one at a time, to simulate the conveyor belt of thymocytes on a background of stationary cells. They are thus able to follow cell lineages, which is interesting because one progenitor can give rise to many progeny.

There are some experimental results in Figures 4,5 and 6. For example, il7 crispant embryos have fewer thymocytes and smaller thymii; but increasing IL-7 availability produces large thymii.

---

## [Referee Report · Reviewer #3 (Public review)]

Summary:

Tsingos et al. seek to advance beyond the current paradigm that proliferation of malignant cells in T-cell acute lymphoblastic leukemia occurs in a cell-autonomous fashion. Using a computational agent-based model and experimental validation, they show instead that cell proliferation also depends on interaction with thymic epithelial cells (TEC) in the thymic niche. One key finding is that a dense TEC network inhibits the proliferation of malignant cells and favors the proliferation of normal cells, whereas a sparse TEC network leads to rapid expansion of malignant thymocytes.

Strengths:

A key strength of this study is that it combines computational modeling using an agent-based model with experimental work. The original modeling and novel experimental work strengthen each other well. In the agent-based model, the authors also tested the effects of varying a few key parameters of cell proliferation.

---

## [Author Response]

The following is the authors’ response to the original reviews

**Public Reviews:**

**Reviewer #1 (Public review):**
Summary:This study uses a cell-based computational model to simulate and study T cell development in the thymus. They initially applied this model to assess the effect of the thymic epithelial cells (TECs) network on thymocyte proliferation and demonstrated that increasing TEC size, density, or protrusions increased the number of thymocytes. They postulated and confirmed that this was due to changes in IL7 signalling and then expanded this work to encompass various environmental and cell-based parameters, including Notch signalling, cell cycle duration, and cell motility. Critical outcomes from the computational model were tested in vivo using medaka fish, such as the role of IL-7 signalling and minimal effect of Notch signalling.Strengths:The strength of the paper is the use of computational modelling to obtain unique insights into the niche parameters that control T cell development, such as the role of TEC architecture, while anchoring those findings with in vivo experiments. I can't comment on the model itself, as I am not an expert in modelling, however, the conclusions of the paper seem to be wellsupported by the model.Weaknesses:One potential issue is that many of the conclusions are drawn from the number of thymocytes, or related parameters such as the thymic size or proliferation of the thymocytes. The study only touches briefly on the influence of the thymic niche on other aspects of thymocyte behaviour, such as their differentiation and death.

We thank the reviewer for this constructive feedback. Indeed, the strength of our approach lies in the close cooperation between modellers and experimentalists. One advantage of the model is its ability to manipulate challenging or even impossible variables, such as TEC dimensions, which cannot be varied experimentally with current tools.

The reviewer rightly pointed out that our validation focuses on comparing cell numbers or organ size as a proxy for cell numbers.

In our previous study (Aghaallaei et al., Science Advances, 2021), we focused more on differentiation and used the computational model to predict how proportions of T-cell sublineages would vary according to different parameter values, including the IL-7 availability. One of the initial inspirations for the focus on proliferation in this manuscript was the observation in this previous work that overexpression of IL-7 in the niche resulted in overproliferation. We also focused on proliferation and organ size because these are more easily measured in experimental conditions with the tools that we have available in medaka, allowing better comparisons to the computational results.

Regarding cell death, our experimental observations do not suggest that it plays a role before the final stages of T cell maturation. Hence, the model also does not include apoptosis before this stage either.

However, we do agree that taking a closer look at the regulation of differentiation and cell death would be an exciting avenue for future study!

Please see our response to author recommendations below for more information on these points. Moreover, to make the model more accessible to non-experts, we have created new schematic figures, which we can be found in the Appendix of the revised manuscript.

**Reviewer #2 (Public review):**
Summary:The authors have worked up a ``virtual thymus' using EPISIM, which has already been published. Attractive features of the computational model are stochasticity, cell-to-cell variability, and spatial heterogeneity. They seek to explore the role of TECs, that release IL-7 which is important in the process of thymocyte division.In the model, ordinary clones have IL7R levels chosen from a distribution, while `lesioned' clones have an IL7R value set to the maximum. The observation is that the lesioned clones are larger families, but the difference is not dramatic. This might be called a cell-intrinsic mechanism. One promising cell-extrinsic mechanism is mentioned: if a lesioned clone happens to be near a source of IL-7 and begins to proliferate, the progeny can crowd out cells of other clones and monopolise the IL-7 source. The effect will be more noticeable if sources are rare, so is seen when the TEC network is sparse.Strengths:Thymic disfunctions are of interest, not least because of T-ALL. New cells are added, one at a time, to simulate the conveyor belt of thymocytes on a background of stationary cells. They are thus able to follow cell lineages, which is interesting because one progenitor can give rise to many progeny.There are some experimental results in Figures 4,5 and 6. For example, il7 crispant embryos have fewer thymocytes and smaller thymii; but increasing IL-7 availability produces large thymii.Weaknesses:On the negative side, like most agent-based models, there are dozens of parameters and assumptions whose values and validity are hard to ascertain.The stated aim is to mimic a 2.5-to-11 day-old medaka thymus, but the constructed model is a geometrical subset that holds about 100 cells at a time in a steady state. The manuscript contains very many figures and lengthy descriptions of simulations run with different parameters values and assumptions. The abstract and conclusion did not help me understand what exactly has been done and learned. No attempt to synthesise observations in any mathematical formula is made.

The reviewer raises several important points to consider when working with mathematical or computational models.

As in many other agent-based models, we agree that our model makes use of many parameters. Many of these parameters summarize multiple steps and are treated as phenomenological, i.e. they do not represent a microscopic event such as the rate of an individual chemical reaction, but more high-level processes such as "rate of differentiation". Realistically, this process should consist of cascades of pathway components that regulate transcription factors.

In the supplementary material of our previous work (Aghaallaei et al., Science Advances, 2021) we provided an in-depth explanation of the mathematical formulation and rationale behind our choices in relation to the available biological data to select assumptions and restrict parameter value ranges. Four parameters that could not be characterized with pre-existing data, but which were crucial to the model's predictions, were studied in detail in that publication. Hence, the submitted manuscript starts with a well-calibrated model that has been tailored for the medaka thymus. The submitted manuscript explores the robustness of the system to lesions, which we conceptualize as alterations in parameter values. We were surprised by how well the model recapitulated the time scales of overproliferation in the thymus of medaka embryos, which further supports the notion that our previous model calibration was successful.

Another important point raised by the reviewer is that the "validity [of parameters and assumptions is] hard to ascertain". We agree, which is precisely the reason why we aim to test the model's predictions through experimentation. Importantly, a model does not need to be perfect to be useful. For example, in the submitted manuscript we observed a discrepancy between model predictions and experimental results that led us to hypothesize negative feedback regulation from the proliferative state to differentiation.

Thus, a major strength of modelling approaches is that they allow to identify erroneous or missing assumptions about the structure of the regulatory interaction network and its parametrization which can advance our scientific understanding of the underlying biology. Using models as an investigative tool is fundamental to the philosophy of systems biology (Kitano, *Science*, 2002), and is what we strive for.

The reviewer rightfully points out that we only represent a geometric subset of the organ. In our preliminary work, we considered representing the full three-dimensional thymus; however, we later simplified our approach, as the organ is a symmetric ellipsoid at this developmental stage. This decision vastly reduced our computational costs, enabling us to explore parameter space more effectively.

Nevertheless, we apologize if the submitted manuscript did not sufficiently emphasize the main insights of the paper, model limitations, and model construction. In the revised manuscript, we have improved the abstract and discussion sections to explicitly highlight the main results and limitations. We have also provided further details of the model's structure and underlying logic in the appendix.

**Reviewer #3 (Public review):**
Summary:Tsingos et al. seek to advance beyond the current paradigm that proliferation of malignant cells in T-cell acute lymphoblastic leukemia occurs in a cell-autonomous fashion. Using a computational agent-based model and experimental validation, they show instead that cell proliferation also depends on interaction with thymic epithelial cells (TEC) in the thymic niche. One key finding is that a dense TEC network inhibits the proliferation of malignant cells and favors the proliferation of normal cells, whereas a sparse TEC network leads to rapid expansion of malignant thymocytes.Strengths:A key strength of this study is that it combines computational modeling using an agent-based model with experimental work. The original modeling and novel experimental work strengthen each other well. In the agent-based model, the authors also tested the effects of varying a few key parameters of cell proliferation.Weaknesses:A minor weakness is that the authors did not conduct a global sensitivity analysis of all parameters in their agent-based model to show that the model is robust to variation, which would demonstrate that their results would still hold under a reasonable level of variation in the model and model parameters. This is a minor point, and such a supporting study would end in an appendix or supplement.

The reviewer highlights the lack of a global sensitivity analysis as a minor weakness.

In our previous work (Aghaallaei et al., Science Advances, 2021), we studied parameters sensitivity for some parameters, while in the submitted manuscript, we extended this exploration to parameters that we expected to be the most meaningful for cell proliferation.

In the revised version of the manuscript, we have included an additional supplementary figure alongside Figure 4 to show the effect of changing parameters in "control" simulations lacking a lesioned clone. These data are also provided in the source data to Figure 4. While this does not constitute an exhaustive exploration of all parameter space, it provides a useful overview of the effect of the studied parameters on thymocyte population size in the absence of lesioned clones.

Response to reviewer recommendations

In the revision, we have improved the manuscript to address the reviewers’ points. The following is an overview of the changes to the manuscript:

• We wrote an extensive Appendix to better explain the model implementation.

• The Abstract was rewritten to improve clarity on what was done and to highlight the main findings.

• Subheadings to paragraphs were rewritten to better emphasize the main findings.

• Font sizes in Figure 2J and Figure 4E were increased to improve readability.

• The spacing of graphical elements in the legend of Figure 4E was improved.

• An error in Figure 5B was corrected (the legend labels had been accidentally swapped).

• A new supplementary figure to Figure 4 shows the sensitivity of clone size in control simulations for a subset of the tested parameter combinations.

• The Conclusion section was rewritten to better highlight limitations of the study and Improve the summary of the main findings.

• Minor wording improvements were done throughout the text to improve readability.

In the following we respond to the reviewers’ individual recommendations.

**Reviewer #1 (Recommendations for the authors):**
I am not an expert in modelling, so I apologise if I missed these points in the manuscript. I am slightly confused about how differentiation and death are included in the model. At the beginning of the results you mention that you model a 5 um slice, is it known which stages of development occur in that section of the thymus?

We thank the reviewer for this question and appreciate the opportunity to clarify. Our virtual thymus is based on the medaka embryonic thymus, which we have extensively characterized using functional analyses and noninvasive in toto imaging (Bajoghli et al., Cell, 2009; Bajoghli et al., J Immunology, 2015; Aghaallaei et al., Science Advances, 2021; Aghaallaei, Eur J Immunology, 2022). These studies allowed us to map thymocyte developmental stages and migratory trajectories within the spatial context of a fully functional medaka thymus (see Figure 7 in Bajoghli et al., J Immunology, 2015).

To simplify the biological system without compromising model fidelity, we chose to simulate a representative 5 µm slice from the ventral half of the thymus. Importantly, the medaka thymus is a symmetric organ (Bajoghli et al., J Immunology 2015), hence this slice captures all key events of T-cell development, including thymus homing, differentiation, proliferation, selection, and egress akin to our in vivo observations (see Figure 7 in Bajoghli et al., 2015 and Figure 7a in Aghaallaei et al., Science Advances, 2021).

Furthermore, our model incorporates the spatial organization of the thymic cortex and medulla by including two types of thymic epithelial cells (TECs): cortical TECs positioned on the outer side, and medullary TECs on the inner side (see Figure Supplement 7 in Aghaallaei et al., *Science Advances*, 2021). Differentiation and cell death are modeled as discrete steps along the developmental trajectory, informed by our in vivo observations.

We apologize to the reviewer if the workings of the model were not sufficiently clear in the original manuscript. To address this, and as also requested by reviewer 2, we provided an extensive Appendix in the revised version of the manuscript that also includes visual summaries of the model logic in the form of intuitive flowcharts.

And is it known, or do you factor in, whether there are changes in the responsiveness of the thymocytes to signals, such as notch and IL7, depending on their state of differentiation?

We have previously examined the roles of IL-7 (Aghaallaei et al., Science Advances, 2021) and Notch1 (Aghaallaei et al., Europ J Immunology, 2022) signaling in the medaka thymus. These studies demonstrated that T cell progenitors are responsive to both IL7 and Notch signaling, whereas more differentiated, non-proliferative thymocytes are unresponsive to IL-7. Our in vivo observations further suggest that mature thymocytes require Notch signaling during the thymic selection process. This appears to be a species-specific phenomenon (Aghaallaei et al., Europ J Immunology, 2022).

In the computational model, we include this state-specific responsiveness by incorporating a dependence on IL-7 and Notch signaling in the cellular decision to commit to the cell cycle (see Appendix Figure 6, and Appendix section X.) and in the decision of differentiating into αβ^+^ or γδ^+^ T cell subtypes (see Appendix Figure 5, and Appendix section IX.). Although the model still calculates pathway signaling activity for thymocytes in the differentiated stage belonging to the αβ^+^ or γδ^+^ subtype, this signaling activity has no downstream consequences for the cells’ behavior in the model.

Note that in the computational model we do not incorporate feedback loops that regulate pathway activity (for example, it could be that thymocytes upregulate the IL7R receptor at some point in their differentiation trajectory – in the absence of speciesspecific knowledge of such regulatory feedbacks, we have chosen not to include any in our model).

And you mention the stages of development are incorporated into the model but the main output that you discuss is thymocyte number or proliferation. It would be interesting to use the model to explore how parameters related to differentiation are changed by, for example, the level of IL7 signalling.

We agree that examining how factors like IL-7 signaling influence thymocyte differentiation is a promising direction for future work. Based on our previous modelling work (Aghaallaei et al., Science Advances, 2021), we expect that increased IL7 availability or sensitivity should result in an increase of cells differentiating into the γδ^+^ T cell subtype. As molecular tools for medaka continue to advance, we anticipate being able to refine and expand the model accordingly.

Moreover, we see strong potential for adapting the current computational framework to model thymopoiesis in other species, such as mouse or human, where stage-specific markers are well characterized. We have now explicitly mentioned this opportunity for future development in the conclusion section of the revised manuscript (see page #26).

It is also mentioned in the description of the model that the cells can die at the end of the development process. However, is death incorporated into the earlier stages of development? For instance, it is possible that when signals, such as a notch, are at low levels the thymocytes at certain stages of development will die.

We thank the reviewer for this comment. In a previous study, we mapped the spatial distribution of apoptotic cells within the medaka thymus and did not observe cell death in the region where ETPs enter the cortical thymus (Bajoghli et al., J Immunology, 2015) and where Notch1 signaling becomes activated (Aghaallaei et al., Europ J Immunology, 2021). Notch mutants exhibit a markedly reduced number of thymocytes, this reduction could be attributed either to impaired thymus homing or increased cell death within the thymus. However, our unpublished data shows that the total number of apoptotic cells in Notch1b-deficient thymus is comparable to their wild-type siblings. In fact, our in vivo observations revealed that the frequency of thymus colonization by progenitors is significantly reduced in the *notch1b* mutant (Aghaallaei et al., J E Immunol., 2021). Based on these in vivo observations, our computational model incorporates cell death only at the end of the thymocyte developmental trajectory. The current model does not consider cell death at earlier stages.

Overall, the manuscript was well-written and the figures were clear and well-presented. A minor point would be that the writing in some of the figures was too small and difficult to read, such as in Figure 4. I also sometimes struggled to find the definition of the acronyms in the figures, for example in Figure 3 it would be helpful if the definitions for D, SD, and SA were given in the figure legend as well as in the figure itself.

We thank the reviewer for the kind words. We have reworked the figures to have larger more readable font sizes and improved figure legends as suggested.

**Reviewer #2 (Recommendations for the authors):**
Suppose the computational results did throw up an important new phenomenon. How might researchers seek to replicate it? If no mathematical relations can be given, can at least the code be made publicly available?

We apologize to the reviewer if the workings of the model were not sufficiently clear in the submitted manuscript. However, we believe there may have been a misunderstanding, and we would like to clarify that both the mathematical formulations and the code used in this study were publicly available in the scientific record at the time of submission.

Specifically, the full source code for the virtual thymus model is hosted in a permanent Zenodo repository (accessible here: https://doi.org/10.5281/zenodo.11656319), which includes:

- Model files and links to source codes for the simulation environment;

- Pre-compiled binary versions of the simulation environment (EPISIM) for both Windows and Linux platforms;

- Detailed documentation, including step-by-step instructions on how to install and use the provided files.

The repository link is cited in the manuscript (see page 38) and in the section “Data and materials availability”.

In addition, the mathematical framework that underpins the computational model has already been published and described in detail in our previous work (Aghaallaei, et al. Science Advances, 2021). In the supplementary material of this publication, we provide extensive documentation of the model, including:

- A 13-page textual explanation of the design rationale;

- 44 equations describing model implementation;

- Parameter choices, partial sensitivity analysis, additional simulations, and supporting data presented in two figures and four tables.

Nonetheless, to improve transparency, we have added an extensive Appendix in the revised version of the manuscript that also includes visual summaries of the model logic in the form of intuitive flowcharts. We hope this clarification and the new provided appendix assures the reviewer that both reproducibility and transparency have been central to our approach.

What about the growth of the animal and its thymus over weeks 2-11?

We thank the reviewer for this insightful question. Indeed, our current computational model does not incorporate thymus growth over time. We decided not to model the dynamic increase in TEC numbers or organ size over time because we wanted to maintain simplicity and computational tractability. Therefore, we assumed a steadystate thymic environment. The model is therefore limited to representing thymopoiesis under homeostatic conditions, as it appears to stabilize by day 11. This is a recognized limitation of the current model. Looking ahead, we plan to develop a more advanced computational framework that incorporates thymic growth and dynamic changes in cellular composition over time. We have now included a brief note on this limitation in the conclusion of the revised manuscript (see page #26).